# Positively charged mineral surfaces promoted the accumulation of organic intermediates at the origin of metabolism

Amir Akbari [1], Bernhard O. Palsson [1,2]*

**1** Department of Bioengineering, University of California, San Diego, California, United States of America, **2** Novo Nordisk Foundation Center for Biosustainability, Technical University of Denmark, Lyngby, Denmark

* palsson@ucsd.edu

**Data Availability Statement:** All the codes, their description, and data used for analysis are provided in Supplementary Information files.

## Abstract

Identifying plausible mechanisms for compartmentalization and accumulation of the organic intermediates of early metabolic cycles in primitive cells has been a major challenge in theories of life's origins. Here, we propose a mechanism, where positive membrane potentials elevate the concentration of the organic intermediates. Positive membrane potentials are generated by positively charged surfaces of protocell membranes due to accumulation of transition metals. We find that (i) positive membrane potentials comparable in magnitude to those of modern cells can increase the concentration of the organic intermediates by several orders of magnitude; (ii) generation of large membrane potentials destabilize ion distributions; (iii) violation of electroneutrality is necessary to induce nonzero membrane potentials; and (iv) violation of electroneutrality enhances osmotic pressure and diminishes reaction efficiency, resulting in an evolutionary driving force for the formation of lipid membranes, specialized ion channels, and active transport systems.

## Author summary

The building blocks of life are constantly synthesized and broken down through concurrent cycles of chemical transformations. Tracing these reactions back 4 billion years to their origins has been a long-standing goal of evolutionary biology. The first metabolic cycles at the origin of life must have overcome several obstacles to spontaneously start and sustain their nonequilibrium states. Notably, maintaining the concentration of organic intermediates at high levels to support their continued operation and evolution would have been challenging in abiotic cells lacking evolutionarily tuned lipid membranes and enzymes. We propose a mechanism, by which the concentrations of organic intermediates are elevated to drive early metabolic cycles in abiotic cells and demonstrate its feasibility in a protocell model from first principles.

**Funding:** This work was funded by the Novo Nordisk Foundation (Grant Number NNF10CC1016517 (BOP)) and the National Institutes of Health (Grant Number GM057089 (BOP)). The funders had no role in study design, data collection and analysis, decision to publish, or preparation of the manuscript.

**Competing interests:** The authors have declared that no competing interests exist.

## Introduction

Metabolism is one of the most fundamental processes of life. At its core, there is a network of chemical transformations conserved across all domains of life [1]. This core network is deeply interconnected with the rest of biochemistry [2], so much so that it is hard to imagine a living system without it. We now have experimental evidence, suggesting that this network has its roots in a geochemical protometabolism originating in deep-sea hydrothermal vents in the Hadean ocean [3, 4]. The metabolism-first hypothesis—one of the main paradigms in prebiotic chemistry—postulates that before the advent of enzymes or genetic codes, geochemical reactions must have emerged spontaneously from simple, inorganic materials and been promoted by naturally occurring catalysts [5–9]. Interestingly, key parts of the core network, including the acetyl-CoA pathway [10, 11], tricarboxylic acid cycle [12–14], glycolysis and pentose phosphate pathway [15, 16], amino-acid and pyrimidine-nucleobase biosynthesis pathways [17, 18], and gluconeogenesis [19] have been reproduced non-enzymatically in laboratory experiments. However, whether a collection of these pathways could have spontaneously arisen and cooperated in prebiotic cell-like structures to form a self-sustaining protometabolic network is yet to be demonstrated.

A central challenge concerning the origins of protometabolism is to show that the organic products of early metabolic pathways could have been produced in sufficiently high concentrations in primitive cells lacking sophisticated machineries of biochemistry, such phospholipid bilayers, membrane proteins, or enzymes. The concentration of these compounds would have been set by a balance between two counteracting mass fluxes, namely membrane transport and reaction rate. The transport rate of organic molecules through porous membranes of primitive cells would have been significantly higher than through lipid bilayers in modern cells, while the rates of nonenzymatic reactions would have been much lower than enzymatic counterparts. Therefore, maintaining the concentration of organic molecules in primitive cells at levels comparable to modern organisms would have been difficult, if not impossible.

In this article, we examine the foregoing challenge—referred to as the concentration problem [20]—more closely by developing a protocell model to understand how the membrane potential and electroneutrality could have elevated the concentration of the organic intermediates of early metabolic reactions in primitive cells. Specifically, we seek to answer (i) whether a positive membrane potential could have developed across protocell membranes under steady state conditions, large enough to concentrate negatively charged organic intermediates of early metabolic networks inside primitive cells, and (ii) if the steady-state solutions could have been stable.

## Results

### Model description

To examine possible mechanisms that could have helped concentrate the organic intermediates of abiotic metabolic cycles, we consider a spherical cell of radius $R_c$ with an ion-permeable porous membrane of thickness $d$, residing in the primitive ocean (Fig 1A). The ocean is assumed electroneutral, only comprising monovalent salts. For simplicity, we only consider two salts with concentrations $C_1^{\text{salt}}$ and $C_2^{\text{salt}}$. Hereafter, we refer to these salts as salt-I and salt-II with $C^{\text{salt}} := C_1^{\text{salt}} + C_2^{\text{salt}}$ the total salt concentration (see "Methods: Protocell Model"). Ions can transfer between the cell and ocean due to a concentration gradient or nonuniform electric-potential field induced by an uneven distribution of cations and anions in the space. Note that the ocean in our idealized model refers to the computational domain outside the cell (see S1 Fig). The main purpose of this model is to determine the membrane potential with respect to

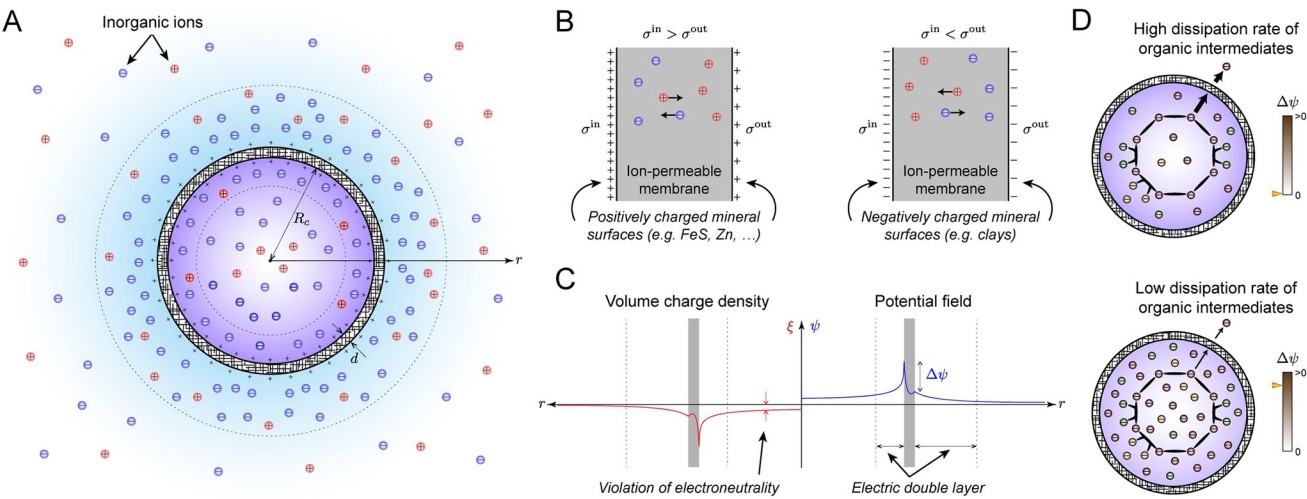

**Fig 1. Proposed protocell model to study the origin of the membrane potential and electroneutrality.** (A) Schematic representation of charge distribution and electric double layers that could have developed inside and around protocells with ion-permeable membranes at the bottom of the primitive ocean. The inner and outer surfaces of the membrane could have been positively charged due to the presence of transition-metal minerals. (B) Two possible mechanisms, through which a nonzero membrane potential could have developed across protocell membranes. The left diagram shows a case, where the inner and outer surfaces of the membrane are positively charged, resulting in a positive membrane potential. The right diagram shows the opposite case, where the inner and outer surfaces are negatively charged, resulting in a negative membrane potential. In both cases, the magnitude of the surface-charge density on the inner surface $\sigma^{in}$ is assumed to be larger than that on the outer surface $\sigma^{out}$. (C) Possible radial profiles for the electric potential $\psi$ and volume-charge density $\xi$ that could lead to a positive membrane potential. Here, the electroneutrality constraint inside the cell and membrane is globally relaxed to achieve a positive membrane potential. (D) Positive membrane potentials lower the dissipation rate of the organic intermediates of early metabolic cycles, thus elevating their concentrations in primitive cells.

the conditions inside and outside the cell. Accordingly, the parameters associated with the ocean in our model, such as $C^{salt}$, should be regarded as characterizing the ambiance of primitive cells if they reside in an environment other than the early ocean.

Our model is concerned with the feasibility of earliest metabolic reactions at the origin of life. The goal is to identify a self-sustaining core reaction network (*e.g.*, a variant of reductive tricarboxylic acid cycle [21]) that could have operated under prebiotic conditions. This network could only rely on simple carbon sources (*i.e.*, $CO_2$ [22] or C2 compounds that could be synthesized abiotically from $CO_2$ [2]) and inorganic compounds (*e.g.*, membrane, energy sources, reducing agents, and catalysts). Prebiotically plausible chemical pathways for the formation of organic compounds besides those in the core network (*e.g.*, single-chain amphiphiles [23] and amino acids [17]) are assumed to have been inactive at the beginning. However, they could have been gradually incorporated into the core network at later stages once their necessary precursors became available in the environment, and they could proceed at high enough rates to ensure the accumulation of their products. As the core network evolved, previously infeasible pathways could have been activated if the network products established autocatalytic feedback loops [24] to amplify these pathways or the concentrations of their intermediates were elevated to levels that furnished nonnegligible reaction rates. However, in this study, we only focus on the feasibility of the core network with regards to the concentration of its intermediates, excluding the foregoing side pathways from the model.

Consistent with the restrictive framework outlined above, the protocell membrane in our model is assumed to be composed of iron sulfides, possibly mixed with other minerals. Vesicles with iron-sulfur membranes could have formed near hydrothermal vents in the Hadean ocean due to the precipitation of $Fe^{2+}$ as iron monosulphide upon mixing with $HS^{-}$-rich alkaline fluids, as has been experimentally demonstrated [25]. Once a self-sustaining core reaction

network had been established in these protocells, the network could have evolved fatty-acid synthesis pathways under appropriate conditions. Interestingly, a wide range of fatty acids could be synthesized through Fischer-Tropsch-type reactions under hydrothermal conditions [26]. Mixtures of fatty acids could then self-assemble into vesicles that remained stable in the presence of divalent ions at high temperatures in extremely saline and alkaline environments [23, 27]. Fatty-acid membranes could have replaced iron-sulfur membranes at later stages of evolution as they could offer a selective advantage by facilitating growth and lipid assimilation [23].

We emphasize that different notions of primitive cells have been introduced in the literature in a variety contexts. To avoid confusion as to what constitutes a primitive cell, we further clarify how it is defined in this work. As previously stated, primitive cell or protocell in this paper refers to an inorganic vesicle with an ion-permeable iron-sulfur membrane. These cells are not equipped with specialized ion channels or transporters, so the only mechanism of membrane transport is unselective, passive diffusion subject to an electric field. This is an important feature of our protocell model: Membranes with a low permeability lower the dissipation rates of organic intermediates and the uptake rates of carbon sources, energy sources, and reducing agents simultaneously, while those with a higher permeability enhance these transport rates. Because of this unselective transport mechanism, neither high nor low membrane permeability would have favored the emergence of first metabolic cycles in such primitive cells. Primitive cells with a membrane made of single-chain amphiphiles have also been considered in previous works. Similar to iron-sulfur membranes, these membranes should also have been ion-permeable to allow the uptake of key nutrients [23, 27, 28]. Therefore, primitive cells with a lipid membrane must have faced the same fundamental challenge as those with an iron-sulfur membrane to facilitate the emergence of first metabolic cycles.

Note that, our goal here is not to computationally exhaust all possible scenarios and physico-chemical systems that could have brought about life on the primitive Earth. Rather, we aim to study a tractable and plausible abiogenic mechanism that possesses the most essential features of primitive cells to gain qualitative insights into the restrictiveness of the constraints on abiotic metabolic evolution through concrete and quantitative analyses. Moreover, the main focus of this study is the role of the membrane potential and electroneutrality in the concentration problem. The feasibility of core reaction networks, operating non-enzymatically in protocells with iron-sulfur membranes and only utilizing inorganic precursors will be examined elsewhere.

The protocell model described above is a simplified version of how primitive cells might have operated in the early ocean. A key simplification concerns the composition of the primitive ocean with regards to the number and valence of ions. As evidenced by leaching studies of oceanic crusts, early oceans were likely more saline than modern seawater, comprising several monovalent and polyvalent ions in high concentrations (*e.g.*, $Na^+$, $K^+$, $Ca^{2+}$, $SO_4^{2-}$) [29]. Nevertheless, by understanding the fundamental constraints that govern electric-potential and charge distributions in our simplified model, we may find clues to possible mechanisms, through which a positive membrane potential could have been achieved in the general case.

In modern cells, there are two main constraints that govern the flow of ions into and out of the cell, namely species mass balance and electroneutrality [30, 31]. Thanks to several specialized ion channels in their ion-impermeable lipid membranes, modern cells can generate local ion gradients in their membranes and a nonzero membrane potential while maintaining electroneutrality on both sides of the membrane [32]. However, in ion-permeable protocell membranes with no specialized ion channels, was it possible to generate a nonzero membrane potential without violating electroneutrality inside the cell? If not, what forms of charge

distribution could have furnished a nonzero membrane potential? Was it necessary for electroneutrality to be violated locally or globally? What mechanism could have underlain electroneutrality violation?

To answer these questions, we replace electroneutrality with Maxwell's first law—a more fundamental constraint governing the interdependence of charge and electric potential. We ascertain the steady-state solutions of Maxwell's first law and species mass balance equations in the cell, membrane, and ocean to identify the key parameters affecting the electric-potential distribution (see "Methods: Governing Equations"). We then determine what parameter values lead to a positive membrane potential through numerical experiments, and, if so, whether electroneutrality can be maintained.

Given that the ocean is electroneutral in our model, one may deduce from Maxwell's first law that inducing a nontrivial electric-potential field in the cell and membrane is not possible without any charge sources. We, thus, consider a scenario, where charged mineral species cover the inner and outer surfaces of the membrane, giving rise to surface charge densities $\sigma^{in}$ and $\sigma^{out}$ (Fig 1B). Transition-metal and clay minerals tend to have positively and negatively charged surfaces, respectively. Here, the idea is that the difference between $\sigma^{in}$ and $\sigma^{out}$ could generate a nontrivial electric-potential field and a nonzero membrane potential as a result. For example, suppose that $|\sigma^{in}| > |\sigma^{out}|$. Then, positively charged mineral surfaces can generate a positive membrane potential (Fig 1C), and negatively charged mineral surfaces can generate a negative membrane potential.

Interestingly, the foregoing scenario is consistent with Wächtershäuser's theory of surface metabolism [6]. In this theory, positively charged mineral surfaces, such as those of divalent transition metals, play a crucial role in facilitating early metabolic reactions because of the strong ionic bonds that form between negatively charged organic molecules and transition metals. In our model, transition metals not only catalyze early metabolic reactions, but also can help concentrate organic molecules inside primitive cells by generating a positive membrane potential if deposited on the surfaces of protocell membranes.

## Membrane surface charges generate a membrane potential

One may deduce from Maxwell's first law that local and global violation of electroneutrality are necessary for the accumulation of charge to generate a nonzero membrane potential. In our model, membrane surface charges cause the violation of electroneutrality and, thus, are generators of the membrane potential. Here, we briefly describe how the membrane potential is developed in our protocell model. Assuming a spherical symmetry, we construct steady-state solutions of Eqs (1) and (2) parametrized with $C^{salt}$ at fixed $R_c$, $\sigma$, and $\sigma_r$ with $\sigma^{in} = \sigma$ and $\sigma^{out} = \sigma_r\sigma$, where $R_c$ is the cell radius, $\sigma$ the inner membrane surface-charge density, and $\sigma_r$ the ratio of the outer to inner membrane surface-charge densities. All other parameters used to obtain the results in this and subsequent sections are summarized in S1 Table.

Steady-state solutions of the electric potential, volume-charge density, and ion concentrations in the cell, membrane, and ocean exhibit a qualitatively similar trend for all values of $C^{salt}$ (S2 Fig). Positively charged membrane surfaces induce a positive electric-potential field in the entire domain ($0 \leq r < \infty$, where $r$ is the radial coordinate) (S2(A) Fig). The resulting membrane potential is also positive, scaling inversely with $C^{salt}$. Electric double layers form around the inner and outer surfaces of the membrane due to surfaces charges. Consequently, negative ions concentrate inside the electric double layers (S2(C) and S2(D) Fig), leading to a net negative volume-charge density throughout the entire domain (S2(B) Fig). We performed this analysis for small and large surface charge densities $\sigma$ and found that the electric potential, volume-charge density, and membrane potential all tend to vanish as $\sigma \rightarrow 0$, as expected. Overall,

membrane surface charges generate a nonzero membrane potential. Electroneutrality is then violated globally in the cell, membrane, and the electric double layer in the ocean as a consequence.

## Trade-off between stability and sensitivity to surface charge could have driven early evolution of membrane potential

The steady-state results presented in the last section indicated that large positive membrane potentials are achievable if mineral surfaces can attain large surface charge densities. Of course, how large a surface-charge density a mineral can attain is dictated by the laws of thermodynamics and depends on several factors, such as the temperature, pressure, and ionic composition of the electrolyte it is exposed to [33]. To better understand if other constraints could have restricted the range of achievable membrane potentials, we systematically studied the steady-state solutions of Eqs (1) and (2) as a function of surface-charge density $\sigma$. We constructed parametric steady state solutions with respect to $\sigma$ at fixed $R_c$, $C^{salt}$, and $\sigma_r$ using branch continuation methods [34] (see "Methods: Constructing Steady-State Solution Branches") and determined stability along the solution branches using linear stability analysis (see "Methods: Stability of Steady-State Solutions").

Steady-state solution branches, represented in scaled membrane potential $\Delta\hat{\psi}$ versus scaled surface-charge density $\hat{\sigma}$ diagrams (see "Methods: Nondimensionalization of Governing Equations" for definition of scaled quantities), exhibit a distinct trend at large (Fig 2A) and small (Fig 2B) cell radii. In general, $\Delta\hat{\psi}$ increases linearly with $\hat{\sigma}$ at small $|\hat{\sigma}|$, asymptotically plateauing at large $|\hat{\sigma}|$. Here, $\Delta\hat{\psi}$ increases with $\hat{\sigma}$ at a faster rate for smaller cells. However, when solution branches are represented with respect to dimensional quantities, the opposite trend is observed; that is, $\Delta\psi$ increases with $\sigma$ at a faster rate for larger cells. Moreover, the linear

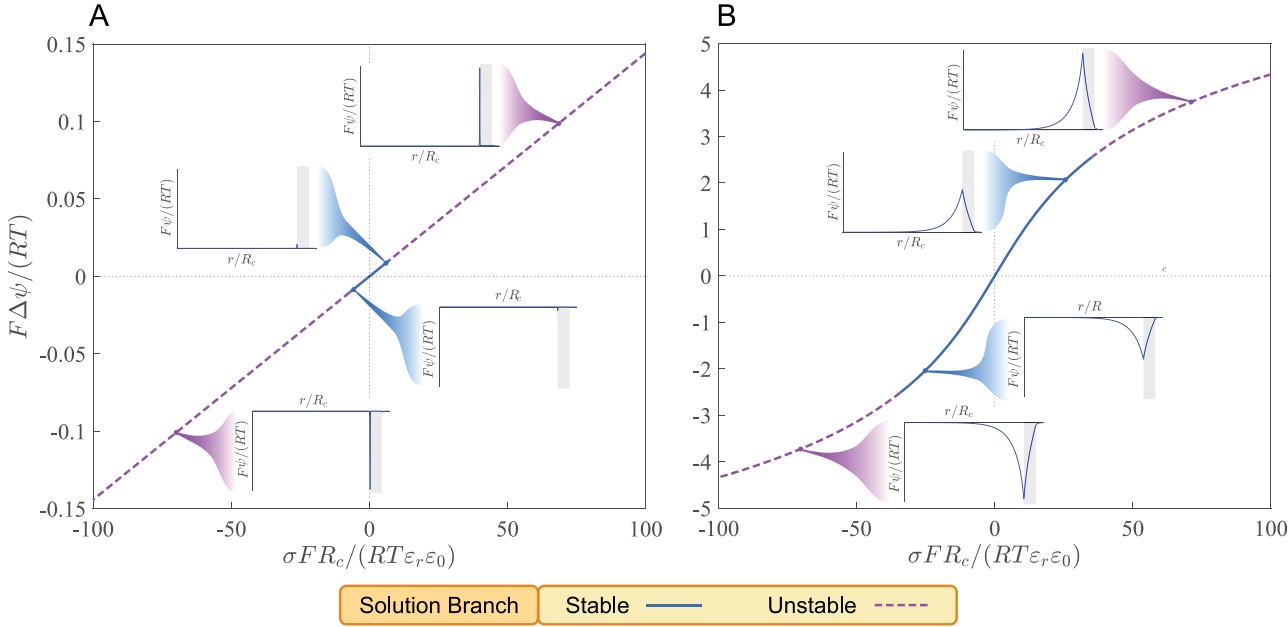

**Fig 2. Steady-state solutions of the membrane potential $\Delta\psi$ parametrically represented with respect to the surface-charge density $\sigma$ at $C^{salt} = 0.1$ M and $\sigma_r = 0.02$.** (A) $R_c = 10^{-6}$ m and (B) $R_c = 10^{-8}$ m. Inset figures show radial profile of the electric potential $\psi$ at selected points along the steady-state solution branches. Shaded areas indicate the position of the membrane along the $r$-axis. Stability analysis was performed for $\vartheta = 0.1$. The tortuosity coefficient $\vartheta$ only affects stability without altering steady-state solutions.

stability analysis shows that, in general, there is a neighborhood around $\hat{\sigma} = 0$, in which steady state solutions are stable. Increasing $\hat{\sigma}$ beyond a critical value results in stability loss. Therefore, unboundedly large membrane potentials cannot be achieved by increasing the membrane surface-charge density arbitrarily, even if allowed by the laws of thermodynamics.

Other parameters besides $\sigma$ could have controlled the development of the membrane potential in primitive cells by altering the steady states and their stability, such as the total salt concentration in the ocean $C^{\text{salt}}$ and the tortuosity coefficient of the membrane $\vartheta$. These characterize the ionic composition of the ocean and microstructural properties of the membrane, respectively. The tortuosity coefficient is defined as the ratio of the effective diffusivity in the membrane and bulk diffusivity for any given ion [35, 36] and measures how much the tortuous microstructure of porous membranes hinders diffusive transport. It only affects the stability of ion distributions in our protocell model, but not the steady-state solutions (see "Methods: Stability of Steady-State Solutions").

We studied how $C^{\text{salt}}$ could influence steady-state solution branches in $\Delta\hat{\psi}$ versus $\hat{\sigma}$ diagrams at large (S3(A) and S3(C) Fig) and small (S3(B) and S3(D) Fig) cell radii by constructing solution branches at fixed values of $C^{\text{salt}}$. We then determined stability along solution branches for $\vartheta = 0.05$ (S3(A) and S3(B) Fig) and $\vartheta = 0.1$ (S3(C) and S3(D) Fig). In general, lowering $C^{\text{salt}}$ increases the sensitivity of $\Delta\hat{\psi}$ to $\hat{\sigma}$. It also reduces the range of $\hat{\sigma}$, in which steady-state solutions are stable. Lowering $\vartheta$ has a similar destabilizing effect by reducing the stable range of $\hat{\sigma}$. Interestingly, the stable regions of $\sigma$ determined in all these case studies are within the experimentally measured ranges of surface charge densities at mineral-water interfaces (see "Methods: Surface Charge of Minerals").

Overall, we observe a trade-off between stability and the sensitivity of the membrane potential $\Delta\psi$ to the surface-charge density $\sigma$, manifesting itself in two distinct ways (Fig 3). The first is connected with the size of the cell. Here, $\Delta\psi$ is more sensitive to $\sigma$ at larger cell radii. However, the range of $\sigma$, in which stable ion distributions can be achieved becomes more restricted in return. Restrictions on $\sigma$ in turn place a constraint on the maximum achievable $\Delta\psi$. The second is related to the composition of the ocean. In this case, a lower $C^{\text{salt}}$ provides $\Delta\psi$ that is more sensitive to $\sigma$, but at the expense of destabilization of ion distributions.

## Electroneutrality could have provided selective advantage by minimizing osmotic pressure and concentration heterogeneity

So far, we focused our discussion on the role that the membrane potential could have played as a barrier, preventing organic molecules, synthesized by earliest metabolic reactions, from dissipating into the ocean. In modern cells, this role has been taken over by lipid bilayer membranes, while the membrane potential together with membrane proteins are involved in membrane bioenergetics [37], pH and metal-ion homeostasis [31], controlling material flow into and out of the cell [32, 38], and stress regulation [30, 39]. As previously stated, a nonzero membrane potential in protocells with ion-permeable membranes would have necessitated the violation of electroneutrality. By contrast, electroneutrality is always maintained in modern cells by controlling ion transfer across the membrane using sophisticated regulatory networks [40, 41]. It may be plausible to surmise that lipid membranes were assimilated by primitive cells from early stages [42] once terpenoid synthesis pathways were incorporated into primordial reaction networks [6]. However, what selective pressures could have driven the evolution of charge distributions towards electroneutrality by selecting for ion-impermeable boundary structures? In the following, we highlight two reasons for why electroneutrality could have been selected for at early stages of evolution, namely (i) the osmotic crisis and (ii) concentration heterogeneity.

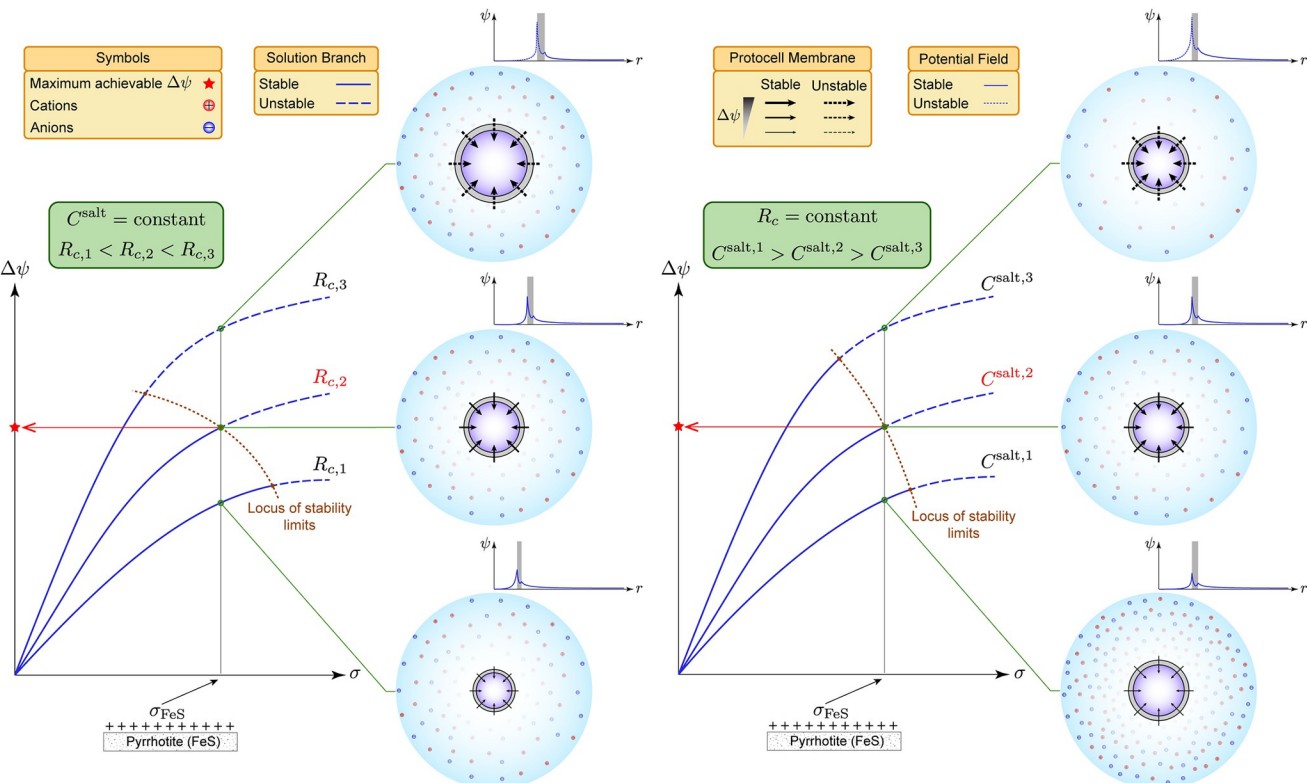

**Fig 3. Trade-off between stability and the sensitivity of the membrane potential $\Delta\psi$ to surface-charge density $\sigma$.** Solution branches for a given tortuosity coefficient $\vartheta$ are qualitatively plotted for several $R_c$ at fixed $C^{\text{salt}}$ (left diagram) and several $C^{\text{salt}}$ at fixed $R_c$ (right diagram). Quantitative solution branches and their stability limits are shown in S3 Fig and S4 Fig, respectively. In both cases, there is a critical $\sigma$ and $\Delta\psi$, beyond which stability is lost. At fixed $C^{\text{salt}}$ (left diagram), larger membrane potentials can be achieved for a given $\sigma$ when $R_c$ is larger, although stability is lost at a smaller $\sigma$. Similarly, at fixed $R_c$ (right diagram), larger membrane potentials can be realized for a given $\sigma$ when $C^{\text{salt}}$ is smaller, but stability is lost at a smaller $\sigma$. Consequently, for a given $\sigma$ corresponding to the constituents of the membrane (*e.g.*, FeS), there is an intermediate $R_c$ at fixed $C^{\text{salt}}$ (highlighted in red in the left diagram) or an intermediate $C^{\text{salt}}$ at fixed $R_c$ (highlighted in red in the right diagram) that furnishes the maximum achievable $\Delta\psi$. Arrows on the membrane point in the direction, in which $\Delta\psi$ exerts force on negative ions.

The osmotic crisis refers to membrane breakup as a result of ion accumulation inside the cell [38]. This universal phenomenon, which applies to primitive and modern cells, occurs due to the uptake of charged cofactors, reducing agents, carbon sources, and energy sources. The elevated ion concentrations in the cell lower the water activity compared to the surrounding water [43]. Consequently, water is driven through the membrane into the cell, causing the membrane to break. To better understand how electroneutrality could have contributed to this process, we considered a simplified case, involving an electrolyte solution with an anion and a cation (see "Methods: Electroneutrality and Structural Stability of Protocells"). We used Pitzer's model [44] to estimate the osmotic coefficient of the solution. Given that the osmotic coefficient is mainly a function of the ionic strength $I$ and total salt concentration $C^{\text{salt}}$ [45], we sought to determine how the osmotic coefficient varies with the net charge of the solution if $I$ and $C^{\text{salt}}$ are maintained constant. We found that, at fixed $I$ and $C^{\text{salt}}$, of all possible ionic compositions, the one yielding an electroneutral solution minimizes the osmotic coefficient. This implies that, primordial reaction networks could have progressed for longer times under electroneutral conditions by taking up all the necessary ingredients from the surroundings without undergoing any catastrophic event. As a result, the likelihood of early metabolism

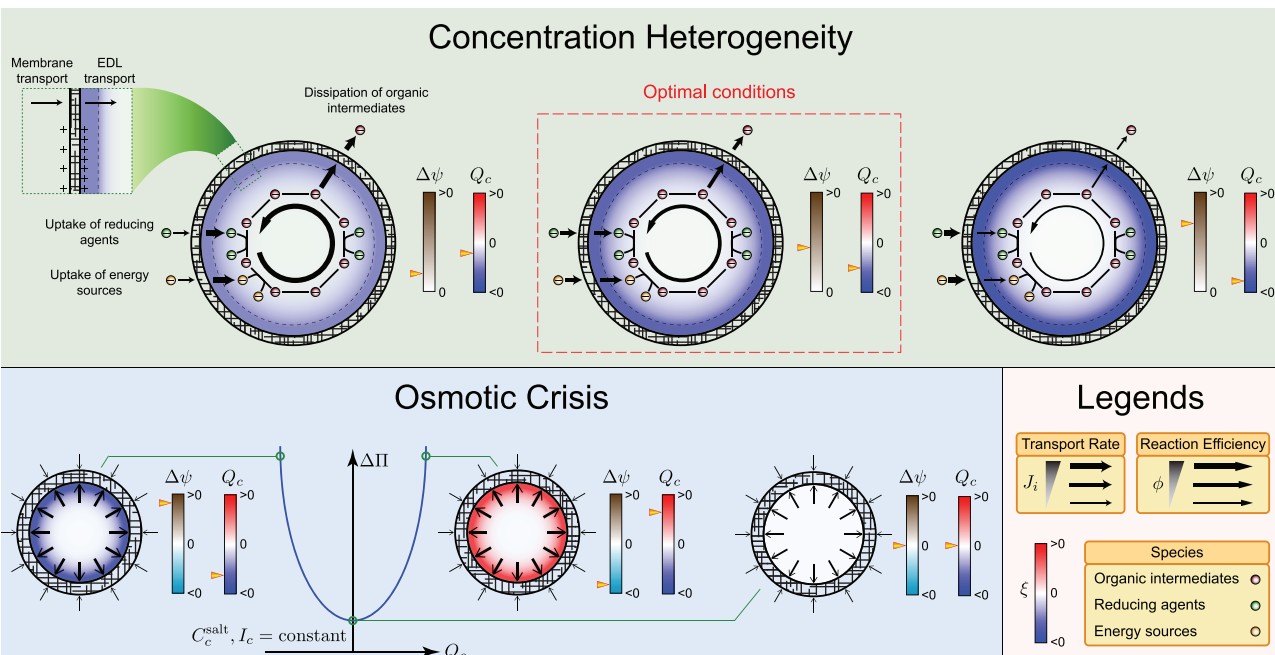

**Fig 4. Selective advantages provided by electroneutrality.** Selective pressures are examined in a protocell model (Fig 1), in which primordial reactions occur. Large positive membrane potentials $\Delta\psi$ increase the transport rate of negatively charged reducing agents and energy sources, while attenuating the dissipation of the organic intermediates of metabolic reactions. They also induce a nonuniform positive electric-potential field, elevating the concentration of anions in the cell. The resulting negative charge-density distribution ($\xi(r) < 0$) leads to heterogeneous concentration distributions for charged reactive species, which in turn diminish reaction efficiencies. Small membrane potentials do not affect the reaction efficiencies significantly. However, they can not alleviate the dissipation of the organic intermediates as effective either. Therefore, parameter values furnishing an intermediate range of positive membrane potentials would have been optimal for the operation of early metabolic cycles. Moreover, if the total salt concentration $C_c^{\text{salt}}$ and ionic strength $I_c$ in the cell are held constant, then violation of electroneutrality ($|Q_c| > 0$ with $Q_c$ the total charge in the cell) increases the osmotic pressure differential $\Delta\Pi$ across the membrane. Hence, electroneutrality could have minimized catastrophic events in primitive cells due to osmotic crisis, promoting the evolution of complex metabolic networks by providing structural stability. Electric double layer is denoted EDL.

incorporating additional steps to synthesize biomolecules of higher complexity could have improved (Fig 4).

Electroneutrality could also have enhanced the efficiency of early metabolic reactions by creating homogeneous concentration distributions inside primitive cells. To clarify this point, consider a scenario, where positively charged membrane surfaces have generated a positive membrane potential in a primitive cell. The positive membrane potential drives negative inorganic ions into the cell, creating a negatively charged shell (*i.e.*, electric double layer) next to the inner surface of the membrane. Next, suppose that a negatively charged organic molecule is consumed by some of the metabolic reactions occurring in the cell. To efficiently utilize this molecule, the cell must maximize its concentration by minimizing its rate of transport into the ocean. This minimization is accomplished by the positive membrane potential. However, the concentration of the molecule is only increased at the inner surface of the membrane since it cannot readily diffuse past the charged shell to mix and react with other molecules. Therefore, organic molecules are nonuniformly distributed inside the cell, diminishing mixing and reaction efficiencies (Fig 4).

To quantify the extent to which nonuniform concentration distributions—referred to as concentration heterogeneity—can reduce apparent reaction rates, we solved the species massbalance equation for a reactive species $B$ consumed according to a first-order rate law in a protocell with a prescribed volume-charge-density distribution to ascertain the radial concentration distribution $C_B(r)$ and its volume average $\langle C_B \rangle$. We then used the reaction efficiency

$\phi_B = \langle C_B \rangle / C_B^{\text{in}}$ as a measure of how much concentration heterogeneity can diminish or enhance apparent reaction rates, where $C_B^{\text{in}}$ is the concentration of $B$ at the inner surface of the membrane (see "Methods: Concentration Heterogeneity and Reaction Efficiency"). We found that deviation from electroneutrality induced by positive membrane potentials could have significantly diminished the efficiency of early metabolic reactions in primitive cells, especially those with smaller cell radii (see S5 Fig).

We also identified two distinct ways, in which concentration heterogeneity can affect apparent reaction rates. The first is due to the local consumption of $B$, always reducing the apparent reaction rate of $B$, irrespective of the background charge distribution arising from the inorganic ions. The second is related to the charged shell discussed above. If the shell and $B$ have opposite charges, the apparent reaction rate is enhanced (S5 Fig, dashed lines), and if they have like charges, the apparent reaction rate is reduced (S5 Fig, solid lines). As it relates to the origin of metabolism, positive membrane potentials could have favored the evolution of early metabolism by minimizing the dissipation of organic intermediates into the ocean. However, they could also have degraded mixing and reaction efficiencies, thereby driving the evolution of ion-impermeable membranes, specialized ion channels, and active transport systems to maintain an electroneural intracellular environment and minimize the interaction of membrane transport and electric double layers.

## Positive membrane potentials could have enhanced the concentration of organic intermediates by several orders of magnitude

Our results suggest that positive membrane potentials that are comparable in magnitude to those observed in modern organisms could have significantly elevated the concentration of negatively charged organic intermediates in primitive cells. For example, we found that $\Delta\psi = 200$ mV can elevate the average concentration of a reactive species $B$ by 10, $10^3$, $10^6$ fold when $z_B = -1, -2, -3$ and $\Lambda_B \ll 1$ in our protocell model with $R_c = 10^{-8}$ m (Fig 5), where $\Lambda_B$ is the Thiele modulus (see "Methods: Concentration Enhancement by Membrane Potential").

Other prebiotic mechanisms for concentration enhancement have been previously examined. For example, surface adsorption on clay minerals in marine and fresh water was suggested to increase the concentration of monomers, facilitating their polymerization at the origin of life [46, 47]. Using numerical simulations, thermophoresis was also shown to enhance the concentration of single nucleotides in hydrothermal pores by $\sim 10^8$ fold [48]. Subsequent laboratory experiments confirmed a $\sim 10^3$ and $\sim 5$ fold accumulation of single nucleotides and fatty acids respectively in microcapillaries using a 30°C temperature gradient, although the Soret effect was largely inhibited at high salt concentrations [49].

Note that, to compare the effectiveness of thermal diffusion forces and positive membrane potentials in addressing the concentration problem, the metric by which concentration enhancement is measured should be carefully evaluated. For example, thermophoresis effects in the foregoing studies were measured by fold increase in local concentrations in contrast to the results presented in Fig 5 based on average concentrations. Concentration enhancement with respect to average concentrations generally underestimates the maximum fold increase in local concentrations in any given system, irrespective of the underlying mechanism. However, average concentrations better characterize reaction-diffusion systems operating in a control volume, such as cellular organisms. Finally, we emphasize that both thermophoresis and membrane potential could have played a role in concentrating the chemical precursors of life in primitive cells. Further investigations are required to determine which mechanism is more congruent with how modern cells overcome the concentration problem and, thus, was favored by natural selection.

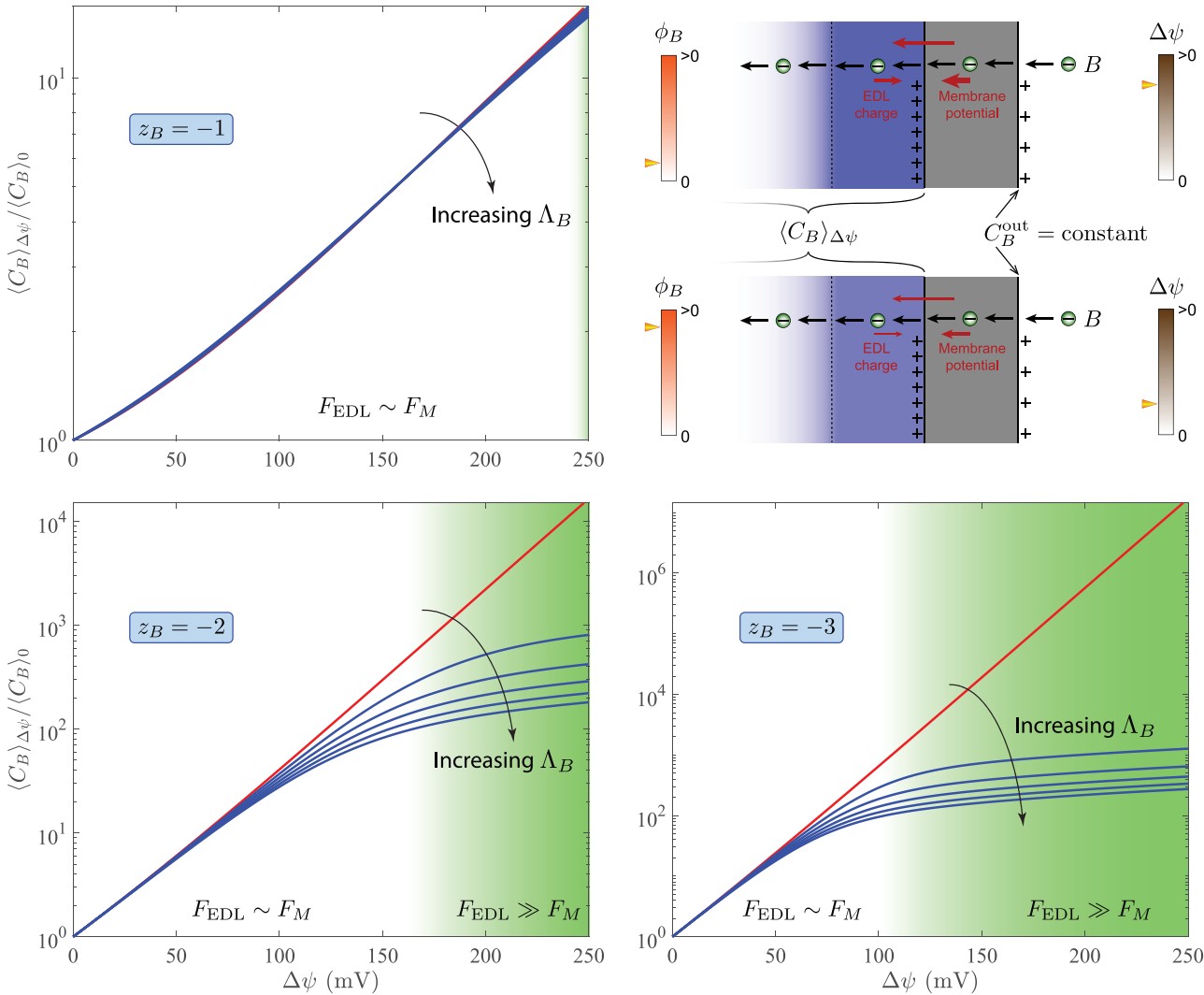

**Fig 5. Enhancement of reactive-species concentrations by the membrane potential.** The membrane potential $\Delta\psi$ and reaction efficiency $\phi_B$ (see "Methods: Concentration Heterogeneity and Reaction Efficiency") for a reactive species $B$ are evaluated at $C^{salt} = 0.1$ M, $\sigma_r = 0.02$, and $R_c = 10^{-8}$ m along a steady-state solution branch parametrized with the surface-charge density $\sigma$. The ratio $\langle C_B \rangle_{\Delta\psi}/\langle C_B \rangle_0$, used as a measure of concentration enhancement, is determined at fixed $C_B^{out}$ for Thiele moduli $\Lambda_B^2 = 0, 0.05, 0.1, 0.15, 0.2, 0.25$ when the valence of $B$ is $z_B = -1, -2, -3$. Red lines represent the nonreactive limit, where $\Lambda_B \rightarrow 0$. Here, $\langle C_B \rangle_{\Delta\psi}$ and $\langle C_B \rangle_0$ denote the average concentration of $B$ in the cell for a given $\Delta\psi$ and when $\Delta\psi = 0$, respectively. Two dimensionless quantities $F_{EDL}$ and $F_M$ determine $\langle C_B \rangle_{\Delta\psi}/\langle C_B \rangle_0$, each measuring the relative strength of the membrane potential driving $B$ into the cell and the electric double layer hindering its diffusion in different ways (see "Methods: Concentration Enhancement by Membrane Potential").

## Conclusions

In this article, we addressed the concentration problem by proposing an abiogenic mechanism, by which the concentration of organic intermediates of early metabolic cycles could have been enhanced without sophisticated macromolecular structures or polymerization machinery, which are believed to have been later products of evolution [50]. In this mechanism, membrane surfaces in primitive cells are assumed to have been positively charged due to the accumulation of transition metals. Then, these charged surfaces could have induced a positive membrane potential, which in turn would have concentrated the organic intermediates of early metabolism in primitive cells. Accordingly, transition metals could have facilitated the

emergence of first metabolic cycles by (i) enhancing the rates of nonenzymatic reactions through catalysis and (ii) alleviating the dissipation of organic intermediates through transport inhibition. We demonstrated the feasibility of this mechanism by developing a protocell model and quantitatively estimating achievable membrane potentials from first principles by solving Maxwell's first law and mass-balance equations. We showed that positive membrane potentials comparable in magnitude to those observed in modern bacteria could have been generated in primitive cells for typical charge densities arising from transition-metal surfaces.

Unlike in our protocell model, the membrane potential in modern cells is often negative [51]. As previously stated, positive membrane potentials would have favored the accumulation of organic intermediates in primitive cells with ion-permeable membranes. Given that the only mechanism of ion transport in these rudimentary membranes was passive diffusion, to achieve a nonzero membrane potential, electroneutrality must have been violated globally inside primitive cells if the early ocean was electroneutral. In contrast, modern cells are equipped with ion-impermeable lipid membranes, efficient enzymes, active transport systems, and proton pumps. In modern cells, the redox energy released by the electron transport chain (ETC) is converted into a chemical potential energy in the form of a pH gradient, which is spent to transport inorganic ions into and out of the cell against their chemical potential gradients through active transport systems [32]. These energy demanding transport mechanisms enable the cell to generate ion-concentration gradients and a nonzero membrane potential across the membrane while maintaining electroneutrality in the cell. Transporters can also transfer organic ions across the membrane selectively in one direction using the pH gradient to alleviate their dissipation. Therefore, modern cell no longer need a positive membrane potential to accumulate the organic products of metabolism. On the other hand, a negative membrane potential in modern cells allows them to store or release a larger fraction of the ETC redox energy per proton when it is translocated across the membrane through proton pumps or the ATP synthase. Thus, negative membrane potentials could have conferred a more robust response to energy demanding stressors and better fitness, which might have been a reason for their evolutionary selection.

To better understand how a positive membrane potential could have developed, we constructed the steady-state solutions of Maxwell's first law and mass-balance equations. We found that positive membrane surface charges could induce a nontrivial electric-potential field and a positive membrane potential. The resulting membrane potential is proportional to the surface-charge density and inversely proportional to the total concentration of nonreactive ions in the primitive ocean. Furthermore, our numerical experiments indicated that violation of electroneutrality inside the cell and membrane is essential to generate a nonzero membrane potential. However, the steady-state results alone did not place any upper limit on the maximum achievable membrane potential.

Thus, we examined the stability of the steady-state solutions using linear stability analysis to identify possible constraints that could have restricted the magnitude of the membrane potential in primitive cells. Our results suggested that, for any given ionic composition of the primitive ocean, there is a critical surface-charge density and membrane potential, beyond which concentration distributions in the cell and membrane are unstable. Moreover, we found that there is a trade-off between this stability bound and the sensitivity of the membrane potential to surface-charge density: Parameter values leading to higher sensitivities result in a smaller range of surface charge densities, for which concentration distributions are stable. Beside destabilization, large surface charge densities could also have induced heterogeneous concentration distributions for organic molecules, cofactors, and energy sources inside primitive cells, adversely affecting the reaction efficiencies of early metabolic cycles. This is yet another reason for why arbitrarily large membrane potentials could not have been achieved.

Lastly, our quantitative analysis revealed that the conditions on the primitive Earth could have been primed for the emergence of first metabolic cycles, perhaps more than previously thought. The feasibility of these cycles in our model relies on the existence of a positive membrane potential. In fact, the concordance between the interconnection of metabolic reactions and the membrane potential in primitive and modern cells is an important feature of our protocell model. It implies that, the operation of the membrane potential and metabolism were deeply intertwined from the outset and continued to persist throughout the evolutionary history of life. Furthermore, our results suggested that, sufficiently large membrane potentials could have been realized for intermediate ranges of surface-charge density, cell size, and ion concentrations in the ocean to support the evolution of stable and self-sustaining metabolic cycles. These ranges may be regarded as constraints exerting selective pressure on the evolution of early metabolism. They would likely have been more restrictive at the beginning and were relaxed once lipid membranes and specialized ion channels had emerged, which in turn would have rendered primitive cells more robust to environmental uncertainties.

Overall, this study provides a strong impetus for further first-principle and quantitative investigations into mechanistic models of first metabolic cycles and their early evolutionary stages, elucidating their transition into self-sustaining and complex biochemical networks. More broadly, our results suggest that the strong interconnections between several cellular processes (*e.g.*, controlled membrane potential and membrane transport, charge balance, ion homeostasis, metabolism) were as essential to primitive cells as they are to extant life. These are fundamental processes that shape many phenotypic characteristics of modern cells, possibly more than currently understood. Therefore, we expect that these fundamental processes will be formalized more systematically in the future for biological systems and incorporated into realistic single-cell models to emerge in the coming decade.

## Materials and methods

### Protocell model

We first describe the protocell model discussed in the main text that we proposed to study the origin of the membrane potential and electroneutrality. The model comprises three regions: (i) Cell, (ii) membrane, and (iii) ocean (S1 Fig). The cell is a sphere of radius $R_c$ enclosed by a porous membrane of thickness $d$, lying at the bottom of the primitive ocean (Fig 1A). Our goal is to determine the conditions, under which a positive membrane potential can develop across the membrane. However, we consider a more general case, where positive and negative membrane potentials can be induced by positively and negatively charged surfaces of the membrane. The inner and outer surfaces of the membrane are assumed to have the same charge with the respective surface-charge densities $\sigma^{in}$ and $\sigma^{out}$. However, the magnitude of the surface-charge density on the inner surface is assumed to be always greater than on the outer surface (Fig 1A). These positively and negatively charged surfaces could have arisen from accumulation of transition-metal and clay minerals, respectively [6]. In our model, the inner and outer surface-charge densities are specified using two parameters according to $\sigma^{in} = \sigma$ and $\sigma^{out} = \sigma_r\sigma$, where the the surface-charge density $\sigma$ and surface-charge density ratio $\sigma_r$ are given parameters. Other fixed parameters of the model that were used to generate the plots in this document and the main text are summarized in S1 Table.

The ocean in our model is assumed to be electroneutral. We further assume that the ionic composition of the ocean arises from a complete dissociation of monovalent salts. For simplicity, we only consider two monovalent salts, we refer to as salt-I and salt-II, which yield equal amounts of their constituent cation and anion in the ocean upon dissociation in water. Let $C_1^{salt}$ and $C_2^{salt}$ denote the concentration of salt-I and salt-II, respectively. Then, the total salt

concentration $C^{\text{salt}} := C_1^{\text{salt}} + C_2^{\text{salt}}$ is half the total ion concentration $C_\infty$ in the ocean due to a complete salt dissociation. Therefore, when $C^{\text{salt}}$ and $r_{12} := C_1^{\text{salt}}/C_2^{\text{salt}}$ are given, the ionic composition of the ocean is fully specified.

$$C_{\infty,i}^+ = C_{\infty,i}^- = C_i^{\text{salt}}, \; i \in \{1,2\},$$

$$C_\infty := \sum_{i \in \{1,2\}} (C_{\infty,i}^+ + C_{\infty,i}^-) = 2C^{\text{salt}},$$

where $C_{\infty,i}^+$ and $C_{\infty,i}^-$ are the concentration of the cation and anion arising from salt $i$. We generally denote the concentration of ions (cations or anions) by $C_{\infty,i}$ without referring to the index of salt, where $i$ here is the index of ions in the system. The composition of the ocean is then imposed as far-field boundary conditions to solve Maxwell's first law and species mass-balance equations in the three computational domains shown in S1 Fig.

## Governing equations

To ascertain the membrane potential for any given set of model parameters, we compute the electric-potential field in the foregoing three computational domains by solving Maxwell's first law

$$\nabla^2 \psi = -\frac{\xi}{\varepsilon_r \varepsilon_0} \tag{1}$$

and species mass-balance equations

$$\frac{\partial C_i}{\partial t} = -\nabla \cdot \mathbf{J}_i + r_i, \; i \in \mathcal{I}, \tag{2}$$

where $\psi$ is the electric potential, $\xi$ volume-charge density, $\varepsilon_r$ relative permittivity of the medium (water in our model), $\varepsilon_0$ vacuum permittivity, $\mathcal{I}$ the index set of all the species involved in the system with $C_i$, $\mathbf{J}_i$, and $r_i$ the concentration, flux vector, and production rate of species $i$. Maxwell's first law describes the relationship between charge and electric-potential distribution in the cell, membrane, and ocean, whether or not electroneutrality holds. The three computational domains, in which to solve Eqs (1) and (2) are represented as

$$
\begin{aligned}
&0 \le r < R_c, && \text{Cell,} \\
&R_c < r < R_c + d, && \text{Membrane,} \\
&R_c + d < r < \infty, && \text{Ocean}
\end{aligned}
\tag{3}
$$

with the boundary conditions

$$\frac{\partial \psi}{\partial r} = 0 \quad \text{at} \quad r = 0, \tag{4a}$$

$$\frac{\partial \psi}{\partial r} = \frac{\sigma^{\text{in}}}{\varepsilon_r \varepsilon_0} \quad \text{at} \quad r = R_c, \tag{4b}$$

$$\psi|_{\text{cell}} = \psi|_{\text{membrane}} \quad \text{at} \quad r = R_c, \tag{4c}$$

$$\psi|_{\text{membrane}} = \psi|_{\text{ocean}} \quad \text{at} \quad r = R_c + d, \tag{4d}$$

$$\frac{\partial \psi}{\partial r} = -\frac{\sigma^{\text{out}}}{\varepsilon_r \varepsilon_0} \quad \text{at} \quad r = R_c + d, \tag{4e}$$

$$\psi \to 0 \quad \text{as} \quad r \to \infty \tag{4f}$$

for the electric potential and

$$\frac{\partial C_i}{\partial r} = 0 \quad \text{at} \quad r = 0, \tag{5a}$$

$$J_i|_{\text{cell}} = J_i|_{\text{membrane}} \quad \text{at} \quad r = R_c, \tag{5b}$$

$$C_i|_{\text{cell}} = C_i|_{\text{membrane}} \quad \text{at} \quad r = R_c, \tag{5c}$$

$$J_i|_{\text{membrane}} = J_i|_{\text{ocean}} \quad \text{at} \quad r = R_c + d, \tag{5d}$$

$$C_i|_{\text{membrane}} = C_i|_{\text{ocean}} \quad \text{at} \quad r = R_c + d, \tag{5e}$$

$$C_i \to C_{\infty,i} \quad \text{as} \quad r \to \infty \tag{5f}$$

for the species concentrations, where $J_i := \mathbf{J}_i \cdot \mathbf{e}_r$ with $\mathbf{e}_r$ the unit vector along the $r$-axis in the spherical coordinate system. The boundary conditions Eqs (4b) and (4e) arise from a charge-balance constraint applied in the electric double layer theory [52, Section 5.3]. It requires that the total charge resulting for the accumulation of the counter ions in the domain exposed to a charged surface counterbalances the total charge of the surface. Applying this constraint to the electric double layers formed in the cell and ocean yields

$$Q_c + \sigma^{\text{in}} A^{\text{in}} = 0, \tag{6}$$

$$Q_o + \sigma^{\text{out}} A^{\text{out}} = 0, \tag{7}$$

where $Q_c$ and $Q_o$ are the total charges accumulated in the cell and ocean with $A^{\text{in}}$ and $A^{\text{out}}$ the areas of the inner and out membrane surfaces. The total charges can then be related to $\frac{\partial \psi}{\partial r}$ using Gauss's law, which is the integral form of Maxwell's first law. For example, Eq (4b) can be derived from Eq (6) in the following way

$$\int_{A^{\text{in}}} \mathbf{E} \cdot \mathbf{n} dA = \frac{Q_c}{\varepsilon_r \varepsilon_0} \Rightarrow \int_{A^{\text{in}}} - \nabla \psi \cdot \mathbf{n} dA = \frac{Q_c}{\varepsilon_r \varepsilon_0}$$

$$\Rightarrow -A^{\text{in}} \left( \frac{\partial \psi}{\partial r} \right)_{r=R_c} = \frac{Q_c}{\varepsilon_r \varepsilon_0} \Rightarrow \left( \frac{\partial \psi}{\partial r} \right)_{r=R_c} = \frac{\sigma^{\text{in}}}{\varepsilon_r \varepsilon_0}$$

with $\mathbf{E}$ the electric field and $\mathbf{n}$ unit outward normal vector to $A^{\text{in}}$. Eq (4c) can be similarly derived from Eq (7).

The volume-charge density in Eq (1) is determined from the concentration of the species in the system

$$\xi = F\sum_{i \in \mathcal{I}} z_i C_i \tag{8}$$

with $F$ the Faraday constant and $z_i$ the valence of species $i$. The flux vector consists of a diffusive and an electric-potential component, which is expressed

$$\mathbf{J}_i = -D_i\left(\nabla C_i + \frac{z_i F}{RT} C_i \nabla \psi\right), \tag{9}$$

where $R$ is the universal gas constant and $T$ temperature.

Before proceeding to the steady-state solutions of Eqs (1) and (2), it is helpful to examine these equations separately for two groups of species. Here, we classify the species involved in our model into a reactive and nonreative group. Reactive species are those that would have participated in early metabolic reactions occurring inside the cell, such as reducing agents, energy sources, and organic molecules. Nonreactive species are the inorganic ions, which would have been present in the primitive ocean.

We note that Eq (2) can be simplified by decoupling the mass balance equation for reactive and nonreactive species. The earliest metabolic reactions are believed to have been catalyzed by naturally occurring minerals at a significantly smaller rate than their enzymatic counterparts [9, 53–55]. As a result, the concentration of reactive compounds produced by these reactions would have been extremely small [54, 56], especially compared to that of inorganic ions in the primitive ocean [29]. Therefore, reactive species could not have significantly contributed to the development of the membrane potential. Accordingly, we neglect their contribution in Eqs (1) and (2), solving Eq (2) only for inorganic ions. These ions are not consumed or produced by any metabolic reactions taking place inside the protocell, so that they attain their steady state much faster than reactive species. Therefore, we solve Eq (2) for $i \in \mathcal{I}_{nrxn}$, approximating the volume-charge density in Eq (1) as

$$\xi \approx F\sum_{\mathcal{I}_{nrxn}} z_i C_i, \tag{10}$$

where $\mathcal{I}_{nrxn}$ the index set of nonreactive species. Using this approximation along with $r_i = 0$ for nonreactive species, one can solve Eqs (1) and (2) for nonreactive species independently of the reactive species. Once the electric potential has been ascertained in this manner, the species mass-balance equations for reactive species can be solved without needing to couple them to Maxwell's first law.

We conclude this section by highlighting the main assumptions used in the remainder of this document to simplify the governing equations and their solutions:

- Governing equations (steady states and transient perturbations) inherit spherical symmetry from the spherical geometry of the protocell model.

- Surface-charge density can vary independently of other model parameters, such as cell radius, membrane thickness, and the ionic composition of the ocean.

The second assumption is only relevant to parametric studies of the steady-state solutions with respect to the surface-charge density $\sigma$. As will be discussed later (see "Constructing Steady-State Solution Branches"), we construct steady-state solution branches with respect to $\sigma$ at fixed $C^{salt}$. This is, of course, a simplifying assumption because when the thermodynamic state of the system is specified (i.e., when the temperature, pressure, and ionic composition are

given), $\sigma$ is determined by the thermodynamic constraints arising from the equilibrium of the charged surface and the electrolyte solution it is subject to [33]. Therefore, in general, $\sigma$ and $C_\infty$ cannot vary independently of one another.

## Nondimensionalization of governing equations

To alleviate computational errors associated with the scaling of the protocell model that arise from the numerical solutions of the governing equations, we introduce the dimensionless quantities

$$\hat{r} := r/R_c, \ \hat{d} := d/R_c, \ \hat{t} := t/\tau, \ \hat{\psi} := F\psi/(RT),$$
$$\hat{\mathbf{J}}_i := R_c \mathbf{J}_i/(D_s C_s), \ \hat{r}_i := R_c^2 r_i/(D_s C_s), \ \hat{C}_i := C_i/C_s, \ \hat{D}_i := D_i/D_s \tag{11}$$

to nondimensionalize these equations. Here, $C_s$, $D_s$, and $\tau := R_c^2/D_s$ are concentration, diffusivity, and time scales. Accordingly, after applying the spherical symmetry assumption, the dimensionless forms of Eqs (1) and (2) are obtained

$$\frac{1}{\hat{r}^2}\frac{\partial}{\partial\hat{r}}\left(\hat{r}^2\frac{\partial\hat{\psi}}{\partial\hat{r}}\right) = -\hat{\xi}, \tag{12}$$

$$\frac{\partial\hat{C}_i}{\partial\hat{t}} = \hat{D}_i\left[\frac{1}{\hat{r}^2}\frac{\partial}{\partial\hat{r}}\left(\hat{r}^2\frac{\partial\hat{C}_i}{\partial\hat{r}}\right) + z_i\frac{\partial\hat{C}_i}{\partial\hat{r}}\frac{\partial\hat{\psi}}{\partial\hat{r}} + \frac{z_i\hat{C}_i}{\hat{r}^2}\frac{\partial}{\partial\hat{r}}\left(\hat{r}^2\frac{\partial\hat{\psi}}{\partial\hat{r}}\right)\right] + \hat{r}_i, \ i \in \mathcal{I}, \tag{13}$$

which are to be solved subject to

$$\frac{\partial\hat{\psi}}{\partial\hat{r}} = 0 \quad \text{at} \quad \hat{r} = 0, \tag{14a}$$

$$\frac{\partial\hat{\psi}}{\partial\hat{r}} = \hat{\sigma}^{\text{in}} \quad \text{at} \quad \hat{r} = 1, \tag{14b}$$

$$\hat{\psi}\big|_{\text{cell}} = \hat{\psi}\big|_{\text{membrane}} \quad \text{at} \quad \hat{r} = 1, \tag{14c}$$

$$\hat{\psi}\big|_{\text{membrane}} = \hat{\psi}\big|_{\text{ocean}} \quad \text{at} \quad \hat{r} = 1 + \hat{d}, \tag{14d}$$

$$\frac{\partial\hat{\psi}}{\partial\hat{r}} = -\hat{\sigma}^{\text{out}} \quad \text{at} \quad \hat{r} = 1 + \hat{d}, \tag{14e}$$

$$\hat{\psi} \to 0 \quad \text{as} \quad \hat{r} \to \infty \tag{14f}$$

for the electric potential and

$$\frac{\partial\hat{C}_i}{\partial\hat{r}} = 0 \quad \text{at} \quad \hat{r} = 0, \tag{15a}$$

$$\hat{J}_i\big|_{\text{cell}} = \hat{J}_i\big|_{\text{membrane}} \quad \text{at} \quad \hat{r} = 1, \tag{15b}$$

$$\hat{C}_i\big|_{\text{cell}} = \hat{C}_i\big|_{\text{membrane}} \quad \text{at} \quad \hat{r} = 1, \tag{15c}$$

$$\hat{J}_i\big|_{\text{membrane}} = \hat{J}_i\big|_{\text{ocean}} \quad \text{at} \quad \hat{r} = 1 + \hat{d}, \tag{15d}$$

$$\hat{C}_i\big|_{\text{membrane}} = \hat{C}_i\big|_{\text{ocean}} \quad \text{at} \quad \hat{r} = 1 + \hat{d}, \tag{15e}$$

$$\hat{C}_i \to \hat{C}_{\infty,i} \quad \text{as} \quad \hat{r} \to \infty \tag{15f}$$

for the species concentrations, where $\hat{J}_i := \hat{\mathbf{J}}_i \cdot \mathbf{e}_r$. Several dimensionless parameters appear in

these equations, the definitions of which are

$$\hat{\xi} := \frac{FR_c^2 \xi}{RT\varepsilon_r\varepsilon_0}, \ \hat{\sigma} := \frac{FR_c\sigma}{RT\varepsilon_r\varepsilon_0}, \ \eta := \frac{F^2 R_c^2 C_s}{RT\varepsilon_r\varepsilon_0}. \tag{16}$$

Note that, throughout this document, the dimensionless forms of all the other concentrations, electric potentials, and surface-charge densities are denoted as the corresponding hatted quantities and defined similarly. Using these dimensionless parameters, Eq (8) is nondimensionalized as

$$\hat{\xi} = \eta \sum_{i \in \mathcal{I}} z_i \hat{C}_i \approx \eta \sum_{i \in \mathcal{I}_{\mathrm{nrxn}}} z_i \hat{C}_i. \tag{17}$$

## Steady state solutions

In this section, we present the steady state solutions of Eqs (12) and (13) for nonreactive species. We approximate the steady-state solutions in the ocean using the Gouy-Chapman theory for simplicity [52] (see "Electric-Potential Field in Ocean from Gouy-Chapman Theory") and compute numerically exact solutions of Maxwell's first law and species mass-balance equations in the cell and membrane. Because there are no sources or sinks for nonreactive species in the cell or membrane, $\hat{J}_i \to 0$ for $i \in \mathcal{I}_{\mathrm{nrxn}}$ as the solutions of Eqs (12) and (13) approach their steady states. This observation allows to simplify the construction of steady-state solutions as demonstrated in the following

$$\hat{J}_i = -\hat{D}_i \left( \frac{d\hat{C}_i}{d\hat{r}} + z_i \hat{C}_i \frac{d\hat{\psi}}{d\hat{r}} \right) = 0 \Rightarrow \frac{d\hat{C}_i}{d\hat{r}} = -z_i \hat{C}_i \frac{d\hat{\psi}}{d\hat{r}}$$

$$\Rightarrow d\ln\hat{C}_i = -z_i d\hat{\psi} \Rightarrow \hat{C}_i(\hat{r}) = \hat{C}_{0,i} \exp\left[ -z_i(\hat{\psi}(\hat{r}) - \hat{\psi}_0) \right],$$

where $\hat{C}_{0,i}$ and $\hat{\psi}_0$ are the concentration of species $i$ and electric potential on one of the boundaries of the computational domain. Note that this simplification does not apply to reactive species, for which $\hat{r}_i \neq 0$, because steady-state fluxes can generally be nonzero.

For the cell and membrane, the boundary of interest is $A^{\mathrm{in}}$ and $A^{\mathrm{out}}$, respectively. The surface concentrations and surface potential on $A^{\mathrm{out}}$, which are ascertained from the Gouy-Chapman theory in the ocean provide the boundary conditions for the membrane. The surface concentrations and surface potential on $A^{\mathrm{in}}$ from the solution of the membrane in turn furnish the boundary conditions for the cell. Once these boundary conditions are substituted in the general expression derived above, the following concentration distributions in the cell and membrane are obtained

$$\hat{C}_i(\hat{r}) = \hat{C}_{\infty,i} \exp\left[ -z_i \hat{\psi}(\hat{r}) \right], \ i \in \mathcal{I}_{\mathrm{nrxn}}, \tag{18}$$

which describe the functional dependence of $\hat{C}_i$ on $\hat{\psi}$ both in the cell and membrane. However, the concentration distributions $\hat{C}_i(\hat{r})$ in the cell and membrane are not the same because the electric-potential field $\hat{\psi}(\hat{r})$ in these domain are different. Substituting Eq (18) in Eq (12) using Eq (17) furnishes

$$\frac{1}{\hat{r}^2} \frac{d}{d\hat{r}} \left( \hat{r}^2 \frac{d\hat{\psi}}{d\hat{r}} \right) = -\eta \sum_{i \in \mathcal{I}_{\mathrm{nrxn}}} z_i \hat{C}_{\infty,i} \exp\left( -z_i \hat{\psi} \right). \tag{19}$$

This form of Maxwell's first law needs not be coupled to the species mass-balance equations

Eq (13) to provide the electric-potential field. We numerically solve this equation using finite-difference methods (see "Numerical Approximation of Steady-State Solutions") subject to the boundary conditions Eqs (14a)–(14b) to compute $\hat{\psi}(\hat{r})$ in the cell and membrane for a given set of model parameters. Once the electric-potential field has been computed, it can be back-substituted in Eq (18) to provide the concentration distributions. S2 Fig represents the results of a case study, showing the radial profiles of the membrane potential, volume-charge density, and ion concentrations that are ascertained by solving Eq (19).

## Electric-potential field in ocean from Gouy-Chapman theory

As previously stated, we approximate the electric-potential field and concentration distribution of ions in the ocean using the Gouy-Chapman theory. This theory provides analytical solutions for $\hat{\psi}(\hat{r})$ and $\hat{C}_i(\hat{r})$ when only monovalent ions are present in an electrolyte solution and the domain is one-dimensional in the Cartesian coordinate system [52]. In this theory, ion concentrations are explicitly expressed as functions of the electric-potential field. The functional form of these expressions is identical to that in Eq (18). The electric-potential field and surface potential are given by

$$\hat{\psi}(\hat{r}) = 4\tanh^{-1}[\tanh(\hat{\psi}^{\text{out}}/4)\exp(-(\hat{r}-1-\hat{d})/\hat{\ell})], \tag{20}$$

$$\hat{\psi}^{\text{out}} = 2\sinh^{-1}(\hat{\ell}\hat{\sigma}^{\text{out}}/2), \tag{21}$$

where $\hat{\ell} := \ell/R_c = 1/\sqrt{2\eta\hat{C}^{\text{salt}}}$ with $\ell$ the Debye length defined as

$$\ell := \sqrt{\frac{RT\varepsilon_r\varepsilon_0}{2C^{\text{salt}}F^2}}. \tag{22}$$

## Numerical approximation of steady-state solutions

We briefly discuss the numerical techniques, with which to solve Eq (19). We discretize Eq (19) over the grid shown in S1 Fig and approximate the first and second derivatives of $\psi$ that arise from its left-hand side using fourth-order finite-difference schemes (see S2 and S3 Tables). Substituting these approximations in Eq (19) yields a nonlinear system of equations

$$\mathbf{A}\hat{\boldsymbol{\psi}} = \mathbf{b}(\hat{\boldsymbol{\psi}}, \hat{\sigma}), \tag{23}$$

where $\hat{\boldsymbol{\psi}} := [\hat{\psi}_1; \hat{\psi}_2; \cdots; \hat{\psi}_{N_c}; \cdots; \hat{\psi}_N] \in \mathbb{R}^{N\times 1}$ is the vector of the electric potentials evaluated at the nodes of the grid shown in S1 Fig. Here, $\mathbf{A} \in \mathbb{R}^{N\times N}$ is a constant matrix comprising the coefficients of the discretization schemes and $\mathbf{b} \in \mathbb{R}^{N\times 1}$ is a variable vector and nonlinear in $\hat{\boldsymbol{\psi}}$. It results from the right-hand side of Eq (19) evaluated at the grid points and the boundary conditions Eqs (14a)–(14d).

We solve Eq (23) iteratively in two steps:

- Pseudo-linear step: The procedure starts from a crude initial guess $\hat{\boldsymbol{\psi}}^0$. At iteration $n$, $\hat{\boldsymbol{\psi}}^n$ is computed by solving the linearized system $\mathbf{A}\hat{\boldsymbol{\psi}}^n = \mathbf{b}(\hat{\boldsymbol{\psi}}^{n-1}, \hat{\sigma})$, where $\hat{\boldsymbol{\psi}}^n$ and $\hat{\boldsymbol{\psi}}^{n-1}$ denote the vector of electric potentials at iterations $n$ and $n-1$, respectively. This step is relatively robust with respect to the initial guess but converges slowly to steady-state solutions. The approximate solution from this step is then used as an initial guess for the next step.

- Newton-Raphson step: Iterations start from the approximate solution furnished by the previous step. At any given iteration, derivative information is used to accelerate convergence towards steady-state solutions. Given the quadratic convergence rate of Newton's method, this step can provide highly accurate solutions with much fewer iterations than the previous step. However, this procedure can also diverge if the initial guess obtained in the previous step does not lie in the convergence region of Newton's method. Therefore, it is not robust with respect to the choice of initial guess.

## Constructing steady-state solution branches

Constructing the solutions of Eq (19) using the computational procedure introduced in the previous section is generally time-consuming, rendering parameter-sweep computations challenging to perform. Therefore, we use Keller's arc-length continuation method [34] to construct steady-state solution branches. The goal is to compute parametric solutions $\hat{\boldsymbol{\psi}}(\hat{\sigma})$ of Eq (23) efficiently by leveraging a predictor-corrector scheme so as to avoid the computational costs associated with repeated execution of the foregoing pseudo-linear step—the computational bottleneck of the procedure outlined in the previous section. Here, we seek $\hat{\boldsymbol{\psi}}(\hat{\sigma})$ as parametric solutions of the problem

$$\mathbf{G}(\hat{\boldsymbol{\psi}}, \hat{\sigma}) := \mathbf{A}\hat{\boldsymbol{\psi}} - \mathbf{b}(\hat{\boldsymbol{\psi}}, \hat{\sigma}) = \mathbf{0} \tag{24}$$

subject to

$$\|\dot{\hat{\boldsymbol{\psi}}}\|_2^2 + \dot{\hat{\sigma}}^2 = 1, \tag{25}$$

where overdot denotes differentiation with respect to the arc-length $s$. The predictor step in branch-continuation methods requires the tangent vector $\dot{\mathbf{x}}$, where $\mathbf{x} := [\hat{\boldsymbol{\psi}}; \hat{\sigma}]$ [34]. The derivatives with respect to $s$ are ascertained by solving

$$\mathcal{G}(\dot{\hat{\boldsymbol{\psi}}}, \dot{\hat{\sigma}}) := \begin{bmatrix} \dfrac{\partial \mathbf{G}}{\partial \hat{\boldsymbol{\psi}}}\dot{\hat{\boldsymbol{\psi}}} + \dfrac{\partial \mathbf{G}}{\partial \hat{\sigma}}\dot{\hat{\sigma}} \\[2ex] \|\dot{\hat{\boldsymbol{\psi}}}\|_2^2 + \dot{\hat{\sigma}}^2 - 1 \end{bmatrix} = \mathbf{0} \tag{26}$$

using Newton's method. Suppose that the vector of steady-state solutions $\mathbf{x}_n$ at the $n$th arc-length step along the solution branch is known with $s_n$ the corresponding arc-length. At this step, the tangent vector $\dot{\mathbf{x}}_n$ can readily be computed by solving Eq (26). The goal now is to compute the solution vector at the next step $\mathbf{x}_{n+1}$ corresponding to $s_{n+1} = s_n + \Delta s$ for a prescribed $\Delta s$. First, an auxiliary function is introduced

$$\mathcal{N}(\hat{\boldsymbol{\psi}}_{n+1}, \hat{\sigma}_{n+1}) := \dot{\hat{\boldsymbol{\psi}}}_n^{\mathrm{T}}(\hat{\boldsymbol{\psi}}_{n+1} - \hat{\boldsymbol{\psi}}_n) + \dot{\hat{\sigma}}_n(\hat{\sigma}_{n+1} - \hat{\sigma}_n) - (s_{n+1} - s_n), \tag{27}$$

which is a linearized version of Eq (25). Next, $\mathbf{x}_{n+1}$ is computed by solving

$$\mathcal{H}(\hat{\boldsymbol{\psi}}_{n+1}, \hat{\sigma}_{n+1}) := \begin{bmatrix} \mathbf{G}(\hat{\boldsymbol{\psi}}_{n+1}, \hat{\sigma}_{n+1}) \\[1.5ex] \mathcal{N}(\hat{\boldsymbol{\psi}}_{n+1}, \hat{\sigma}_{n+1}) \end{bmatrix} = \mathbf{0} \tag{28}$$

using a predictor-corrector scheme. In the predictor step, the tangent vector $\dot{\mathbf{x}}_n$ is used to construct an initial guess for $\mathbf{x}_{n+1}$ as follows

$$\mathbf{x}_{n+1}^{\circ} = \mathbf{x}_n + \dot{\mathbf{x}}_n \Delta s,$$

which is accurate to first order in $\Delta s$. In the corrector step, $\mathbf{x}_{n+1}$ is computed to high accuracy by solving Eq (28) using Newton's method with $\mathbf{x}_{n+1}^{\circ}$ as an initial guess. A key advantage of predictor-corrector approaches is that, the initial guess generated in the predictor step usually lies in the convergence region of Newton's method even for moderately sized $\Delta s$. Moreover, constructing the initial guess in the predictor step is computationally much less costly than the pseudo-linear step discussed in the previous section. Therefore, parameter-sweep computations can be performed much more efficiently using branch-continuation methods than the procedure introduced in the previous section, if it were to be executed at all points along the solution branch.

## Stability of steady-state solutions

We determine the stability of steady-state solutions using linear stability analysis. Let $\hat{\psi}^0(\hat{r})$ and $\hat{C}_i^0(\hat{r})$ denote the steady-state solutions of Eqs (12) and (13). Upon perturbations, the time-varying solutions of Eqs (12) and (13) can be expressed as

$$
\begin{aligned}
\hat{\psi}(\hat{t}, \hat{r}) &= \hat{\psi}^0(\hat{r}) + \hat{\psi}'(\hat{t}, \hat{r}), \\
\hat{C}_i(\hat{t}, \hat{r}) &= \hat{C}_i^0(\hat{r}) + \hat{C}_i'(\hat{t}, \hat{r}),
\end{aligned}
\tag{29}
$$

where $\hat{\psi}'$ and $\hat{C}_i'$ are infinitesimal perturbations induced in the protocell by fluctuations in environmental conditions. To simplify the analysis, we assume that these fluctuations can only destabilize the concentration distributions in the cell and membrane without affecting the steady state of the ocean. Accordingly, the concentration and electric-potential boundary conditions on $A^{\text{out}}$ are not influenced by these perturbations. We further assume that instabilities are mainly caused by concentration perturbations, neglecting the disturbances that they can induce in the electric-potential field (*i.e.*, $\hat{\psi}'(\hat{t}, \hat{r}) = 0$). Thus, we consider the following perturbation *ansatz*

$$
\hat{C}_i'(\hat{t}, \hat{r}) = \exp(\lambda_i \hat{t}) \rho_i(\hat{r})
\tag{30}
$$

with $\lambda_i$ the eigenvalue characterizing the dynamics of species $i$. Substituting Eqs (29) and (30) in Eq (13), taking into account all the assumptions discussed above and neglecting the second- and higher-order terms in $\hat{C}_i'$, we arrive at

$$
\frac{d^2 \rho_i}{d\hat{r}^2} + \left( \frac{2}{\hat{r}} + z_i \frac{d\hat{\psi}^0}{d\hat{r}} \right) \frac{d\rho_i}{d\hat{r}} - \left( z_i \hat{\xi}^0 + \frac{\lambda_i}{\mathcal{D}_i} \right) \rho_i = 0, \quad
\begin{cases}
0 < \hat{r} < 1, \\
1 < \hat{r} < 1 + \hat{d}
\end{cases}
\tag{31}
$$

subject to

$$
\frac{d\rho_i}{d\hat{r}} = 0 \quad \text{at} \quad \hat{r} = 0,
\tag{32a}
$$

$$
\left[ \frac{d\rho_i}{d\hat{r}} + z_i \frac{d\hat{\psi}^0}{d\hat{r}} \rho_i \right]_{\text{cell}} = \vartheta \left[ \frac{d\rho_i}{d\hat{r}} + z_i \frac{d\hat{\psi}^0}{d\hat{r}} \rho_i \right]_{\text{membrane}} \quad \text{at} \quad \hat{r} = 1,
\tag{32b}
$$

$$
\rho_i|_{\text{cell}} = \rho_i|_{\text{membrane}} \quad \text{at} \quad \hat{r} = 1,
\tag{32c}
$$

$$
\rho_i = 0 \quad \text{at} \quad \hat{r} = 1 + \hat{d},
\tag{32d}
$$

where $\mathcal{D}_i$ is a diffusivity coefficient defined as

$$\mathcal{D}_i := \begin{cases} D_i, & 0 \leq \hat{r} < 1, \\ D_i^{\text{eff}}, & 1 < \hat{r} < 1 + \hat{d} \end{cases} \tag{33}$$

with $D_i$ the the bulk diffusivity of species $i$ in water and $D_i^{\text{eff}}$ its effective diffusivity in the membrane. In our protocell model, the membrane is assumed to have a porous structure made of minerals, the effective diffusivity of which can be expressed as $D_i^{\text{eff}} = \vartheta D_i$ with $\vartheta$ the tortuosity coefficient [35, 36]. Using these definitions, the boundary condition Eq (32b) is derived from Eq (15b).

Eq (31) can be recast into the following Sturm-Liouville form for each $i$

$$\frac{\mathrm{d}}{\mathrm{d}\hat{r}}\left[p(\hat{r})\frac{\mathrm{d}\rho_i}{\mathrm{d}\hat{r}}\right] - q(\hat{r})\rho_i = \lambda_i w(\hat{r})\rho_i, \tag{34}$$

where

$$p(\hat{r}) := \mu_i(\hat{r}), \tag{35a}$$

$$q(\hat{r}) := z_i\mu_i(\hat{r})\hat{\bar{\xi}}^0(\hat{r}), \tag{35b}$$

$$w(\hat{r}) := \mu_i(\hat{r})/\mathcal{D}_i \tag{35c}$$

with

$$\mu_i(\hat{r}) := \hat{r}^2 \exp\left[z_i\hat{\psi}^0(\hat{r})\right]. \tag{36}$$

The Sturm-Liouville problem Eq (34) is called regular if it is subject to some variants of homogeneous Robin boundary conditions, $p(\hat{r}), w(\hat{r}) > 0$, and $p(\hat{r})$, $\mathrm{d}p(\hat{r})/\mathrm{d}\hat{r}$, $q(\hat{r})$, and $w(\hat{r})$ are continuous on $[0, 1 + \hat{d}]$ [57]. The following properties of regular Sturm-Liouville problems are of particular relevance to linear stability analysis [58]:

- Eigenvalues are discrete and real.

- Eigenvalues are bounded from above.

- Eigenfunctions form a complete orthogonal basis for an $L_2$ Hilbert space.

Moreover, the solutions of a regular Sturm-Liouville problem are continuously differentiable [58]. However, the Sturm-Liouville problem that arise from Eq (31) is not regular. Firstly, the boundary condition at $\hat{r} = 0$ is singular because $p(0) = 0$. Secondly, $p(\hat{r})$, $q(\hat{r})$, and $w(\hat{r})$ are nonsmooth at $\hat{r} = 1$. Nonetheless, modern treatments of the Sturm-Liouville theory allows these functions to satisfy more relaxed conditions, such that the foregoing three properties still hold. Accordingly, it suffices for $1/p(\hat{r})$, $q(\hat{r})$, and $w(\hat{r})$ to be locally Lebesgue integrable—a condition satisfied by Eq (35). Although, the solutions (*i.e.*, eigenfunctions) may satisfy weaker smoothness and continuity properties. This generalization followed from the important finding that Hilbert function spaces can be decomposed into mutually orthogonal singular and absolutely continuous subspaces for self-adjoint operators (see [57] for more details).

Given the properties of Sturm-Liouville problems discussed above, it suffices to show that the maximum eigenvalue of Eq (31) subject to the boundary conditions Eq (32a)–(32d) is negative to prove that a steady-state solution is stable. We, thus, solve Eq (31) numerically using finite-difference methods following a similar procedure as discussed before (see "Numerical Approximation of Steady-State Solutions"). Discretization of Eq (31) subject to Eq (32a)–(32d)

results in the following systems of equations

$$\mathbf{R}_i(\hat{\psi}^0, \lambda_i) = \mathbf{0}, \ i \in \mathcal{I}_{\mathrm{nrxn}}, \tag{37}$$

which have a nontrivial solution if and only if $\mathbf{R}_i(\hat{\psi}^0, \lambda_i)$ are singular. The determinant can be used as a measure of how far a matrix is from being singular, the application of which leads to the following condition

$$\det \mathbf{R}_i(\hat{\psi}^0, \lambda_i) = 0, \ i \in \mathcal{I}_{\mathrm{nrxn}}. \tag{38}$$

Determining matrix singularity from Eq (38) is computationally expensive. Therefore, we consider an alternative condition for $\mathbf{R}_i(\hat{\psi}^0, \lambda_i)$ to be singular by requiring its minimum singular value to vanish

$$\sigma_{\min} \mathbf{R}_i(\hat{\psi}^0, \lambda_i) = 0, \ i \in \mathcal{I}_{\mathrm{nrxn}}, \tag{39}$$

where $\sigma_{\min}$ is an operator returning the minimum singular value of $\mathbf{R}_i$ (not to be confused with the surface-charge densities introduced in previous sections). Solving Eq (39) is computationally less expensive than solving Eq (38) since the singular values of $\mathbf{R}_i$ can be efficiently computed by leveraging its sparsity structure. Once Eq (39) has been solved, the stability of steady-state solutions can be determined by the sign of $\lambda_i^{\max}$ for all $i \in \mathcal{I}_{\mathrm{nrxn}}$, where $\lambda_i^{\max}$ is the maximum eigenvalue of $\mathbf{R}_i(\hat{\psi}^0, \lambda_i)$ (see S3 and S4 Figs).

## Concentration heterogeneity and reaction efficiency

So far, we restricted our analysis to electric-potential fields that could have been indued by nonuniform distributions of inorganic ions of the primitive ocean (*i.e.*, nonreactive species). These nontrivial potential fields could have affected the operation and evolution of early metabolic cycles. Many of the organic molecules, reducing agents, and energy sources participating in these metabolic reactions would have been negatively charged, the transport of which would have been altered by the background electric-potential field arising from the inorganic ions. Therefore, the resulting concentration distribution of these reactive species would have been heterogeneous, which in turn would have adversely affected the efficiency of early metabolic reactions. In this section, we study this phenomenon by solving diffusion-reaction mass-balance equations for reactive species that are subject to a prescribed background electric-potential field in the cell.

Unlike nonreactive species, the steady-state fluxes of reactive species are generally nonzero. Hence, the solution strategy that we previously discussed (see "Steayd-State Solutions") for nonreactive species is not applicable here. The concentration distribution of reactive species is also not described by Eq (18). Therefore, we numerically construct the steady-state solutions of Eqs (12) and (13) for reactive species. The goal is to quantify the extent to which reaction efficiencies are affected by the background electric-potential field in the cell. In the following, we first describe a quantitative measure of reaction efficiencies.

Suppose that $B$ is a negatively charged reactive species and a substrate consumed by metabolic reactions taking place in the cell, which is to be imported from the ocean into the cell (*e.g.*, reducing agents or energy sources). To maximize the rate of metabolic reactions, the concentration of $B$ in the cell must be maintained at the highest possible level. A positive membrane potential can enhance the transport rate of $B$ from the ocean to the cell, increasing its concentration at the inner surface of the membrane $C_B^{\mathrm{in}}$, which could potentially enhance the rates of metabolic reactions. However, the overall consumption rate of $B$ in the entire volume of the cell depends on its concentration distribution. A uniform distribution $C_B(r) = C_B^{\mathrm{in}}$

would ensure that $B$ is maximally utilized by the metabolic reactions in the cell. However, uniform concentration distributions are generally not achievable due to local consumption of $B$ and the background electric-potential field. To quantify how concentration distributions can affect the overall consumption rate of $B$, we study a macroscopic description of its reaction-diffusion mass balance by examining the integral form of Eq (13). Integrating Eq (13) over the volume of the cell for $B$ and applying the divergence theorem result in

$$\frac{d\langle\hat{C}_B\rangle}{d\hat{t}} = -3\hat{J}_B + \langle\hat{r}_B\rangle, \tag{40}$$

where

$$\langle\hat{C}_B\rangle := \frac{1}{\hat{V}}\int_{\hat{V}}\hat{C}_B d\hat{V}, \tag{41a}$$

$$\langle\hat{r}_B\rangle := \frac{1}{\hat{V}}\int_{\hat{V}}\hat{r}_B d\hat{V} \tag{41b}$$

with $\hat{V} := V/R_c^3 = 4\pi/3$. We refer to $\langle\hat{r}_B\rangle$ as the apparent production rate of $B$, which is a negative number here because $B$ is consumed by metabolic reactions. The rate, at which the products of the reactions that $B$ participates in are generated is proportional to $-\langle\hat{r}_B\rangle$. Clearly, concentration distributions that maximize $-\langle\hat{r}_B\rangle$ favor the progress of these metabolic reactions. To quantify the extent to which concentration distributions can enhance the overall rates of these metabolic reactions, we compare $\langle\hat{r}_B\rangle$ for a given concentration distribution to what it would be if $B$ was uniformly distributed in the cell—the ideal distribution that maximizes its utilization. Accordingly, we define the following reaction efficiency for the consumption of $B$

$$\phi_B := \frac{\langle\hat{r}_B\rangle_{\hat{C}_B(\hat{r})}}{\langle\hat{r}_B\rangle_{\hat{C}_B^{in}}}. \tag{42}$$

The relationship between $\langle r_B\rangle$ and $\langle C_B\rangle$ for nonlinear rate laws is not straightforward. Hence, we assume that $B$ is consumed in the cell according to the first-order rate law $r_B = -kC_B$ to simplify the analysis. The apparent reaction rate from this rate law is also linear with respect to the average concentration, that is $\langle r_B\rangle = -k\langle C_B\rangle$. Accordingly, the reaction efficiency with respect to this rate law is

$$\phi_B = \frac{\langle C_B\rangle}{C_B^{in}}, \tag{43}$$

which we use as a measure of how much concentration heterogeneity can diminish or enhance the rates of metabolic reactions consuming $B$. Note that $0 < \phi_B \leq 1$ only when $B$ is negatively charged. However, when $B$ is positively charged, $\phi_B$ can be greater than one.

Next, we compute the steady-state solutions of Eq (13) for $B$ using finite-difference techniques along the solution branches shown in Fig 2. These solution branches represent the steady states of the electric-potential field induced by the inorganic ions of the ocean parametrized with the surface-charge density. We express the dimensionless reaction rate $\hat{r}_B$ with respect to the Thiele modulus $\Lambda_B$ and construct the concentration distribution of $B$ in the cell that arise from the first-order rate law $r_B = -kC_B$ by solving

$$\frac{1}{\hat{r}^2}\frac{d}{d\hat{r}}\left(\hat{r}^2\frac{d\hat{C}_B}{d\hat{r}}\right) + z_B\frac{d\hat{C}_B}{d\hat{r}}\frac{d\hat{\psi}}{d\hat{r}} + \left[\frac{z_B}{\hat{r}^2}\frac{d}{d\hat{r}}\left(\hat{r}^2\frac{d\hat{\psi}}{d\hat{r}}\right) - \Lambda_B^2\right]\hat{C}_B = 0 \tag{44}$$

subject to

$$\frac{\mathrm{d}\hat{C}_B}{\mathrm{d}\hat{r}} = 0 \quad \text{at} \quad \hat{r} = 0, \quad (45\text{a})$$

$$\hat{C}_B = \hat{C}_B^{\text{in}} \quad \text{at} \quad \hat{r} = 1, \quad (45\text{b})$$

where

$$\Lambda_B := \sqrt{\frac{R_c^2 k}{D_B}} \qquad (46)$$

is the Thiele modulus [59].

Once the concentration distribution of $B$ has been determined, we compute the reaction efficiency from Eq (43). As expected, the reaction efficiency for positively charged species is higher than for negatively charged ones because these ions must diffuse through a negatively charged medium to participate in metabolic reactions that occur in the cell (S5 Fig). Higher surface-charge densities cause more negative ions to accumulate in the cell, amplifying this effect. The reaction efficiency is always less than one for negatively charged species. However, it can exceed one for positively charged species if the surface-charge density is large enough. Note that, the background charge induced by the inorganic ions of the ocean is not the only parameter affecting the reaction efficiency. The local consumption of reactants can also result in heterogeneous concentration distributions, irrespective of the background charge. This effect is more conspicuous in the limit $\sigma \to 0$. Even though the entire volume of the cell is electroneutral in this limit, the reaction efficiency can be less than one (S5 Fig). Note also that, diminished reaction efficiencies as a result of local mass sinks is more pronounced at larger Thiele moduli.

## Electroneutrality and structural stability of protocells

Electroneutrality is often treated as a fundamental law governing the state of electrolyte systems [45]. It is also regarded as a fundamental constraint that biological systems are subject to [31, 41]. As such, it is believed to underlie regulatory responses to several stress conditions [31, 40]. However, as discussed in the main text, violation of electroneutrality is essential for the mechanism that we proposed to promote the evolution of early metabolic cycles in primitive cells that lack lipid membranes and enzymes. To corroborate this mechanism, we examine the possibility that electroneutrality was not a fundamental constraint at the earliest stages of evolution. The goal is to understand whether electroneutrality could have resulted from the evolution of lipid membranes, specialized ion channels, and active transport systems selected for to minimize catastrophic events due to osmotic crisis. From this perspective, electroneutrality is an emergent property of evolving systems, self-optimizing towards a state of maximal structural stability through natural selection.

Here, we quantitatively examine the relationship between osmitic crisis and electroneutrality through a simplified case study. We consider an electrolyte, comprising a cation $M^+$ and an anion $X^-$. Our objective is to determine how the osmotic pressure arising from this system varies with the charge density of the solution and assess if the minimum osmotic pressure is attained when the solution is electroneutral. There are two main variables that determine the osmotic coefficient of an electrolyte, namely the (molal) ionic strength $I_m$ and the total (molal) concentration $m$ of the solution [45]. The idea is to focus solely on the role of electroneutrality and identify the (molal) charge density $\xi_m$ that minimizes the osmotic coefficient $\varphi$ at fixed $I_m$ and $m$.

The state of single electrolyte systems $MX$ at fixed temperature and pressure is specified by two variables, namely the molal concentrations $m_M$ and $m_X$. Hence, specifying $\xi_m$, $I_m$, and $m$ for a general single electrolyte system overdetermines its state. However, for a monovalent electrolyte, such as the one we considered here, $I_m = m/2$. Thus, if the ionic strength and total concentration are fixed, the charge density can still vary freely without causing an inconsistent degree of freedom. We begin by stating the virial expansion of the excess Gibbs energy of mixing

$$\frac{G^{\text{ex}}}{n_w RT} = f(I_m) + \sum_{ij} \omega_{ij} m_i m_j + \sum_{ijk} \omega_{ijk} m_i m_j m_k + \cdots, \tag{47}$$

which is the basis of Pitzer's model, where $n_w$ is the mass of water in kg [45]. The first term in Eq (47) captures long-range electrostatic forces and the rest capture medium- and short-range interactions among ions. To simplify the analysis, we omit the terms corresponding to interactions among three or more ions as they are negligible in most cases [60]. Differentiating the excess Gibbs energy with respect to $n_w$ yields the osmotic coefficient

$$\varphi - 1 = \frac{I_m \mathrm{d}f/\mathrm{d}I_m - f}{m} + \frac{1}{m} \sum_{ij} u_{ij} m_i m_j + \mathcal{O}\left(\frac{m_i^3}{m}\right), \tag{48}$$

where

$$\xi_m := z_M m_M + z_X m_X,$$

$$I_m := (z_M^2 m_M + z_X^2 m_X)/2,$$

$$u_{ij} := \omega_{ij} + I_m \mathrm{d}\omega_{ij}/\mathrm{d}I_m.$$

To make the analysis more concrete, we consider a case, where the electrolyte $MX$ is contained in a protocell lying at the bottom of the primitive ocean, such as that shown in Fig 1. The osmotic pressure differential across the cell membrane is derived from Eq (48)

$$\frac{\Delta\Pi}{\hat{\rho}_w RT} = m + I_m \mathrm{d}f/\mathrm{d}I_m - f + \sum_{ij} u_{ij} m_i m_j - \frac{\Pi^{\text{out}}}{\hat{\rho}_w RT}, \tag{49}$$

where $\Delta\Pi := \Pi^{\text{in}} - \Pi^{\text{out}}$ is the osmotic pressure differential, $\Pi^{\text{in}}$ pressure in the cell, $\Pi^{\text{out}}$ pressure in the ocean, and $\hat{\rho}_w$ reduced water density (see [31] for definition and detailed discussion). We assume that the thermodynamic state of the ocean is specified, so that the last term in Eq (49) is a constant. In this equation, only the term corresponding to the second virial coefficient of Eq (47) on the right-hand side depends on $\xi_m$. Therefore, it is the only variable term, with respect to which $\Delta\Pi$ is minimized. Accordingly, we seek $\xi_m$ that minimizes

$$\theta := \sum_{ij} u_{ij} m_i m_j. \tag{50}$$

For the monovalent electrolyte $MX$, the concentration of ions can be expressed with respect to $m$ and $\xi_m$ as

$$m_M = \frac{m + \xi_m}{2},$$

$$m_X = \frac{m - \xi_m}{2}. \tag{51}$$

Substituting Eq (51) in Eq (50) and nondimensionalizing the resulting terms yields

$$\bar{\theta} = a_2 \bar{\bar{\xi}}_m^2 + a_1 \bar{\bar{\xi}}_m + a_0 \tag{52}$$

where

$$a_0 := \bar{u}_{MM} + 2\bar{u}_{MX} + \bar{u}_{XX}, \tag{53a}$$

$$a_1 := 2(\bar{u}_{MM} - \bar{u}_{XX}), \tag{53b}$$

$$a_2 := \bar{u}_{MM} - 2\bar{u}_{MX} + \bar{u}_{XX} \tag{53c}$$

with $\bar{\theta} := 4\theta/m$, $\bar{u}_{ij} := u_{ij}m$, and $\bar{\bar{\xi}} := \xi/m$.

Next, we leverage the properties of the coefficients $a_i$ in Eq (53) that can be deduced from experimental observations. First, thermodynamic mixing properties are not significantly affected by the diagonal second virial coefficients for most electrolyte systems, so that $\omega_{MM} = 0$ and $\omega_{XX} = 0$ [61], which in turn results in $\bar{u}_{MM} = 0$, $\bar{u}_{XX} = 0$, and $a_1 = 0$. Second, $a_0 < 0$ for a wide range of dilute electrolytes ($I_m \lesssim 0.25$ mol/kg-w) [45], from which it follows that $a_2 > 0$. One can deduce from these empirical properties and the functional form of $\bar{\theta}$ in Eq (52) that $\bar{\theta}$ has a minimum, and it is attained at $\bar{\bar{\xi}}_m = 0$.

Finally, we emphasize that the results presented in this section for a single monovalent electrolyte cannot be regarded as a rigorous proof. Nevertheless, they support the hypothesis that electroneutral systems are subject to minimal osmotic stress. More analyses are required to generalize these results to mixed electrolytes with polyvalent ions. Given the role of electrostatic forces in short- and medium-range ion-ion interactions, it may be plausible to assume that the nonlinear proportional relationship between the osmotic pressure differential $\Delta\Pi$ and absolute charge $|\xi_m|$ is generalizable to more complex electrolyte systems. However, whether the minimum osmotic pressure is always attained exactly at $\bar{\bar{\xi}}_m = 0$, regardless of the molecular characteristics of the ions involved, warrants further investigations.

## Surface charge of minerals

The mechanism we introduced in this paper to generate positive membrane potentials hinges on porous membranes with positively charged surfaces. In this mechanism, the membrane potential is larger if the surface-charge density on the inner surface is larger than on the outer surface of the membrane. From our case studies, we found that $\Delta\psi \sim 100$ mV can be achieved in small protocells with radius $R_c \sim 10^{-8}$ m for $\Delta\sigma \sim 0.1$ C/m$^2$ (see S4 Fig), where $\Delta\sigma := \sigma^{\text{in}} - \sigma^{\text{out}}$ is the surface-charge-density differential across the membrane. In this section, we describe a specific scenario based on experimental measurements of the surface-charge density for how such positive surface charge could have been realized in primitive cells.

Solid surfaces, such as those of minerals, can adsorb or desorb ions (usually $H^+$, $OH^-$, or other ions that may be present in the system) from or to water when exposed to an aqueous phase. As a result, these surfaces may acquire a surface charge. When the temperature, pressure, and ionic composition of the electrolyte that mineral surfaces are subject to are specified, the surface-charge density is a function of pH. At fixed temperature and pressure, the functional form of $\sigma(\text{pH})$ for each mineral depends on its constituents and the composition of the electrolyte, which can typically be represented by a monotonically decreasing function, such as those shown in S6 Fig [62]. However, non-monotonic $\sigma(\text{pH})$ have also been observed (for example, see Fig 6 in [63]). Nevertheless, we only focus on minerals that exhibit a monotonic $\sigma$(pH) in this section.

The monotonicity of $\sigma(\text{pH})$ implies that $|\sigma|$ attains its maximum in the alkaline and acidic limits. Accordingly, surface-charge densities for most minerals are observed in the range $\sigma(\text{pH} = 1) < \sigma < \sigma(\text{pH} = 14)$. For example, surfaces charge densities in the range $-0.4 < \sigma < 0.4$ C/m$^2$ for synthetic and natural ferrous minerals, which are relevant to the conditions on the primitive Earth [6, 64], have been reported [65, 66] with a similar $\sigma(\text{pH})$ to those shown in S6 Fig. This range generally agrees in order of magnitude with the stable ranges of $\sigma$ that we ascertained in our case studies (see S3 and S4 Figs).

Alkaline hydrothermal vents are considered to be one of the likely environments, in which life could have originated [9, 67]. In these environments, $CO_2$-rich acidic ocean water (pH $\approx$ 5) could have interfaced with alkaline hydrothermal fluids (pH $\approx$ 9), providing suitable conditions for the first metabolic reactions to emerge [64]. Recent experimental evidence indicates that such pH gradients could have provided sufficient energy to drive the thermodynamically unfavorable carbon-fixation steps of early metabolism under prebiotic conditions [68]. Here, we suggest that the pH gradient between hydrothermal fluids and the primitive ocean could also have generated sufficiently large surface-charge-density differentials across protocell membranes ($\Delta\sigma \sim 0.1$ C/m$^2$).

To clarify the point raised above, consider a protocell, residing at an interface between a hydrothermal vent and ocean (S6 Fig). Suppose that acidic and alkaline fluids flow into the cell, neutralizing each other, such that pH $\approx$ 7 in the cell. The neutral pH in primitive cells would have been optimal for the emergence of surface metabolism at the origin of life [6]. The difference between $\sigma^{\text{in}}$ at pH = 7 and $\sigma^{\text{out}}$ at pH = 9 may be large or small, depending on the constituent minerals of the membrane. Mineral-I and Mineral-II in S6 Fig represent two hypothetical minerals that could generate small and large $\Delta\sigma$, respectively. A key difference between the functional form of $\sigma(\text{pH})$ for these minerals is in the point of zero charge (PZC) [62]. The PCZ occurs at pH $\approx$ 7.5 for Mineral-I. Thus, $\sigma^{\text{in}}$ and $\sigma^{\text{out}}$ are both small because of the small slope of $\sigma(\text{pH})$ at the PCZ, so that $\Delta\sigma \approx 0$ C/m$^2$. However, for Mineral-II, the PCZ occurs at pH $\approx$ 9.3. As a result, $\sigma^{\text{in}}$ and $\sigma^{\text{out}}$ are both positive with $\Delta\sigma \approx 0.1$ C/m$^2$. Therefore, the scenario we described in Fig 1B for generating positive membrane potentials could have been realized for protocell membranes made of Mineral-II. Interestingly, experimental measurements of the surface-charge density of mineral-water interfaces indicate that the functional from of $\sigma(\text{pH})$ for several transition-metal sulfides (e.g., $Ni_3S_2$ and ZnS) is similar to that of Mineral-II in S6 Fig with the PZC in the alkaline range [63, 69].

## Concentration enhancement by membrane potential

In the main text, we discussed how positive membrane potentials could have enhanced the concentration of the organic intermediates of early metabolic cycles in protocells lacking lipid membranes and enzymes. To quantify the extend, to which a positive membrane potential could have elevated the concentration of a negatively charged reactive species $B$, we define

$$\varphi_B := \langle C_B \rangle_{\Delta\psi} / \langle C_B \rangle_0, \tag{54}$$

where $\langle C_B \rangle_{\Delta\psi}$ and $\langle C_B \rangle_0$ denote the average concentration of $B$ in the cell for the given $\Delta\psi$ and when $\Delta\psi = 0$, respectively. We use $\varphi_B$ as a measure of concentration enhancement in our protocell model. We first determine $\langle C_B \rangle_{\Delta\psi}$ from the steady-state solution of Eq (40)

$$-3\hat{J}_B + \langle \hat{r}_B \rangle = 0, \tag{55}$$

assuming the first-order rate law $r_B = -kC_B$, as before. The membrane-potential-dependent

flux $\hat{J}_B$ is estimated [32]

$$\hat{J}_B = \frac{\vartheta \hat{D}_B}{\hat{d}} z_B \Delta \hat{\psi} \frac{\hat{C}_B^{\text{in}} \exp\left(z_B \Delta \hat{\psi}\right) - \hat{C}_B^{\text{out}}}{\exp\left(z_B \Delta \hat{\psi}\right) - 1}. \tag{56}$$

Note that, this approximate expression is derived by assuming a linear potential field in a flat membrane of thickness $d$. However, as we observed previously (see S2(A) Fig), solving Eq (19) in the spherical membrane of the protocell model in Fig 1A does not generally furnish a linear $\psi(r)$. Nevertheless, the compact form of Eq (56) allows us to obtain an analytical expression for $\langle C_B \rangle$. Substituting $C_B^{\text{in}} = \langle C_B \rangle / \phi_B$ in Eq (56) furnishes

$$\frac{\langle C_B \rangle}{C_B^{\text{out}}} = \left[\frac{1}{F_{\text{EDL}}} + \frac{\gamma}{F_{\text{M}}}\right]^{-1} \tag{57}$$

with

$$F_{\text{EDL}} \; := \; \frac{\phi_B}{\exp\left(z_B \Delta \hat{\psi}\right)}, \tag{58a}$$

$$F_{\text{M}} \; := \; \frac{z_B \Delta \hat{\psi}}{\exp\left(z_B \Delta \hat{\psi}\right) - 1}, \tag{58b}$$

$$\gamma \; := \; \frac{\hat{d} \Lambda_B^2}{3\vartheta}. \tag{58c}$$

Eq (57) furnishes $\langle C_B \rangle$ as a function of $\Delta \psi$. Accordingly, $\langle C_B \rangle_{\Delta \psi} := \langle C_B \rangle(\Delta \psi)$ and $\langle C_B \rangle_0 := \langle C_B \rangle(0)$ can be readily ascertained from Eq (57a) to determine $\varphi_B$. Note also that $\varphi_B$ in Eq (58a) is a function of $\Delta \psi$ and $\Lambda_B$ (see "Concentration Heterogeneity and Reaction Efficiency").

   We conclude this section by highlighting the physical interpretation of the two dimensionless quantities $F_{\text{EDL}}$ and $F_{\text{M}}/\gamma$. The first measures the relative strength of concentration heterogeneity (due to both local consumption of $B$ and background electric-potential field) diminishing $\langle r_B \rangle$ by lowering $\varphi_B$ and the membrane potential enhancing $\langle r_B \rangle$ by increasing $-J_B$. The second also measures the relative strength of concentration heterogeneity (due to local consumption of $B$) diminishing $\langle r_B \rangle$ by increasing $\Lambda_B$ and the membrane potential enhancing $\langle r_B \rangle$ by increasing $-J_B$. When $\Delta \hat{\psi} \ll 1$, $F_{\text{EDL}} \sim F_{\text{M}}$, whereas $F_{\text{EDL}} \gg F_{\text{M}}$, when $\Delta \hat{\psi} \gg 1$. As a result, $\varphi_B$ exhibits a transition from a log-linear regime when $\Delta \hat{\psi} \ll 1$ to an asymptotic regime when $\Delta \hat{\psi} \gg 1$ (see Fig 5).

## Supporting information

**S1 Fig. Computational domains in the protocell model of life's origins described in Fig 1 and the grid used to discretize the governing equations.** The model comprises three computational domains, namely the cell, membrane, and ocean. Maxwell's first law and species mass-balance equations are solved in the cell and membrane to ascertain the electric-potential field and concentration distributions. However, the surface potential on the outer surface of the membrane, electric-potential field, and concentration distributions in the ocean are approximated by the Gouy-Chapman theory [52, Section 5.3].
(PDF)

**S2 Fig. Steady-state solutions of species mass-balance and Maxwell's first equations at $R_c = 10^{-7}$ m, $\sigma = 0.01$ C/m², and $\sigma_r = 0.2$.** (A) Electric potential, (B) volume-charge density, (C) concentration of cations and anions associated with salt-I, and (D) concentration of cations

and anions associated with salt-II. Here, $C_{\infty,i}$ denotes the far-field concentration of ion $i$ in the ocean, arising from dissociation of the respect salt in water. Shaded areas indicate the position of the membrane along the $r$-axis. The scale on the $r$-axis, where the ocean lies, is stretched by a factor 20 to better show radial profiles.
(PDF)

**S3 Fig. Stability along steady-state solution branches of $\Delta\psi$ parametrized with respect to the surface-charge density $\sigma$ at $\sigma_r$ = 0.02.** (A) $R_c = 10^{-6}$ m and $\vartheta = 0.05$, (B) $R_c = 10^{-8}$ m and $\vartheta = 0.05$, (C) $R_c = 10^{-6}$ m and $\vartheta = 0.1$, and (D) $R_c = 10^{-8}$ m and $\vartheta = 0.1$. Colorbars indicate the value of $C^{\text{salt}} = C_{\infty}/2$ that corresponds to each curve in (A)–(D). The tortuosity coefficient $\vartheta$ only affects stability without altering steady-state solutions. Solid and dashed lines represent stable and unstable steady-state solutions, respectively.
(PDF)

**S4 Fig. Stability limits along steady-state solution branches of S3 Fig in the positive orthant.** (A) surface-charge density, at which stability is lost. (B) Membrane potential, at which stability is lost.
(PDF)

**S5 Fig. Average concentration of the cation (dashed lines) and anion (solid lines) arising from the dissociation of a monovalent salt inside the cell at $C^{\text{salt}}$ = 0.1 M, $\Lambda_B^2 = 0, 10, 20, 30, 40, 50$.** (A) $R_c = 10^{-6}$ m and (B) $R_c = 10^{-8}$ m. Each Thiele modulus $\Lambda_B$ corresponds to a solid-dashed curve pair, increasing along the direction indicated by the arrows. Red lines represent the nonreactive limit, where $\Lambda_B \rightarrow 0$.
(PDF)

**S6 Fig. Typical surface-charge density of minerals measured as a function of pH using potentiometric-conductometric titration experiments [62, Chapter 1].** A pH gradient across the membrane of the protocell model shown in Fig 1 is assumed to cause a surface charge-density differential between the inner and outer surfaces of the membrane. The protocell resides near a hydrothermal vent at an interface between two fluids with pH $\approx$ 9 and pH $\approx$ 5. The fluids flow into the cell from the alkaline vent and acidic ocean. The acidic and alkaline fluids neutralize into water, such that pH $\approx$ 7 in the cell. Mineral-I and Mineral-II are two hypothetical minerals that exhibit different functional forms for $\sigma(\text{pH})$. For the given pH gradient between the hydrothermal vent and ocean, the surface-charge densities formed on the inner and outer surfaces of a membrane made of Mineral-I are almost identical. However, a large surface-charge-density differential can be generated across a membrane made of Mineral-II, such that $\sigma^{\text{in}}$ and $\sigma^{\text{out}}$ are both positive, similarly to the scenario described in the left diagram of Fig 1B.
(PDF)

**S1 File. Custom codes used to perform all the computations and generate the results and figures in main text and supplementary information.**
(ZIP)

**S1 Table. Parameters used for all the case studies presented in the main text and supplementary information.**
(PDF)

**S2 Table. Forth-order finite-difference schemes to approximate the first derivative of a function $f(\hat{r})$ on the grid shown in S1 Fig.** Expressions in the Scheme column are derivatives

evaluated at $\hat{r}_j$, where $f'_j := f'(\hat{r}_j)$. Note that $\Delta\hat{r} := \Delta\hat{r}_c$ when $1 \le j \le N_c$ and $\Delta\hat{r} := \Delta\hat{r}_m$ when $N_c + 1 \le j \le N$ (see S1 Fig).
(PDF)

**S3 Table. Forth-order finite-difference schemes to approximate the second derivative of a function $f(\hat{r})$ on the grid shown in S1 Fig.** Expressions in the Scheme column are derivatives evaluated at $\hat{r}_j$, where $f''_j := f''(\hat{r}_j)$. Note that $\Delta\hat{r} := \Delta\hat{r}_c$ when $1 \le j \le N_c$ and $\Delta\hat{r} := \Delta\hat{r}_m$ when $N_c + 1 \le j \le N$ (see S1 Fig).
(PDF)

## Acknowledgments

We thank Daniel Zielinski and Marc Abrams for their comments on the manuscript.

## Author Contributions

**Conceptualization:** Amir Akbari, Bernhard O. Palsson.

**Formal analysis:** Amir Akbari.

**Funding acquisition:** Bernhard O. Palsson.

**Investigation:** Amir Akbari.

**Methodology:** Amir Akbari.

**Resources:** Bernhard O. Palsson.

**Software:** Amir Akbari.

**Supervision:** Bernhard O. Palsson.

**Validation:** Amir Akbari.

**Visualization:** Amir Akbari.

**Writing – original draft:** Amir Akbari, Bernhard O. Palsson.

**Writing – review & editing:** Amir Akbari, Bernhard O. Palsson.

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
