## [Decision Letter · Decision Letter 0]

7 Jun 2022

Dear Dr Palsson,

Thank you very much for submitting your manuscript "Positively charged mineral surfaces promoted the accumulation of organic intermediates at the origin of life" for consideration at PLOS Computational Biology.

As with all papers reviewed by the journal, your manuscript was reviewed by members of the editorial board and by two independent reviewers. Apologies again for the lengthy time it took us to secure reviews for your manuscript. I believe it’s a combination of the pandemic aftermath, that makes it generally more difficult to recruit reviewers, and the topic of the manuscript, that appeals to a specialized field of researchers that are interested in the origins of metabolism.

Both reviewers are highly knowledgeable in the origin of metabolism field, and both have published (prominent, mostly experimental) results about non-enzymatic metabolism-like reactions. As you can extract from their comments, your study did appeal to one reviewer (an organic chemist), and not appeal to the other reviewer (the biochemist), who was essentially lost.

I recommend a major revision for your manuscript, I would like however ask you not only to address the specific comments of the reviewers, but also, for revising the manuscript to make it more appealing and more accessible to the biochemists and the biologists – which after all, are the key audience you need to convince if your paper is to create impact.

Two Editorial suggestions in this regard:  

In the first part of the discussion, the manuscript talks about the dispute that existed in the literature about whether protometabolic #cycles# would be chemically feasible. You quote some well-known, but 15+ year old ‘opinion -type’ papers --  but there is a significant amount of more recent, experimental, literature, that actually discovered that Fe(II) and other transition metals catalyze metabolism like reactions, that mimic glycolysis, pentose phosphate pathway and Krebs cycle. It might help your manuscript to set the scene around experimentally proven facts, rather then speculative review-type pieces by Orgel and co, that divided the filed a long time ago.  Specifically, it may indeed appeal to the biologists if they are reminded that there is experimental evidence that glycolysis and the TCA -like reactions are catalyzed by metal irons. Btw, does it really need to be ‘cycles’ (in the metabolic network sense), or do reaction sequences suffice?

Second, what might help is the focus the manuscript not on the overly broad topic of the origin of life in general, but around the specific problem about the origins of the metabolic pathways, and the metabolic network topology. There is an argument that these processes are connected, but the manuscript might sound much more tangible if it would discuss specifically the origin of the extant metabolic network structure (which may or may not be the origin of life - its not essential to go into this debate in this specific manuscript - we know as a fact that the metabolic network evolved at some point (early on), it does not really matter for this manuscript if it was the first step).

Thank you again for your submission. We cannot make any decision about publication until we have seen the revised manuscript and your response to the reviewers' comments. Your revised manuscript is also likely to be sent to reviewers for further evaluation.We hope that our editorial process has been constructive so far, and we welcome your feedback at any time. Please don't hesitate to contact us if you have any questions or comments. 

Sincerely,

Markus Ralser

Guest Editor

PLOS Computational Biology

Kiran Patil

Deputy Editor

PLOS Computational Biology

Reviewer's Responses to Questions

**Comments to the Authors:**

Reviewer #1: Akbari and Palsson propose that positive membrane potentials caused by the accumulation of transition metals on the surface could have elevated the concentration of organic intermediates in a protometabolism at the earliest stages of the origin of life. They propose a mathematical relationship between the membrane potential at steady-state and the surface -charge density, including radial profiles. They show a trade-off between the stability of the membrane and the sensitivity of the membrane potential to surface-charge density. It is concluded that electroneutrality would be advantageous to a protocell with a protometabolism because osmotic pressure and concentration heterogeneity would be minimized.

The general idea of this paper is similar to the idea previously proposed by Wächtershäuser in his theory of surface metabolism (ref. 6), as acknowledged by the authors. However, Wächtershäuser’s proposal was based purely on intuition, whereas the model treated here is mathematical and quantitative in nature. This puts this work on a much more solid level. In general, I found this paper well written, well-reasoned and well referenced. I could recommend it for publication.

I have a few comments that the authors might want to consider addressing, at their discretion.

-Some of the ideas regarding the “concentration problem” in protometabolism have also been expressed with regards to the polymerization of monomers. In a 1951 book cited below, Bernal suggested that concentrations of the products, which would be an absolute necessity for evolution, could be attained by adsorption on very fine clay deposits in marine and fresh water.

Book: J. D. Bernal: The Physical Basis of Life. Routledge and Kegan Paul, London 1951.

An experimental example of this proposal was later shown by Katchalsky.

Paecht-Horowitz, M.; Berger, J.; Katchalsky, A. Prebiotic Synthesis of Polypeptides by Heterogeneous Polycondensation of Amino-Acid Adenylates. Nature 1970, 228 (5272), 636-639. https://doi.org/10.1038/228636a0.

-Although Wächtershäuser is often credited for the igniting the field of protometabolism, one of the first was Granick.

Granick, S. SPECULATIONS ON THE ORIGINS AND EVOLUTION OF PHOTOSYNTHESIS. Ann. NY Acad Sci. 1957, 69, 292-308. https://doi.org/10.1111/j.1749-6632.1957.tb49665.x.

-On pg. 3, the authors state “metabolic reactions could not have proceeded fast enough to synthesize necessary precursors for the evolution of complex reaction networks without enzymes or genetic replication.” I don’t know on what basis this claim can be made. Very few experimentalists have worked on protometabolism, so this conclusion is difficult to reach in the absence of experimentation.

--On pg. 3 “primitive cells lacking lipid membranes”. Perhaps the authors should define exactly what they mean by primitive cells. Lots of readers would assume this involves a lipid membrane, so precisely defining what is meant here could be useful.

Regarding the impact of elevated metabolite concentrations on catalysis, in general a problem in catalysis is the inhibition of the catalyst by binding to the product. If the membrane potential increases metabolite concentrations too much, could product-inhibition become more problematic than the problem of low concentrations? Where might this trade-off happen?

-Pg. 8, “the total salt concentration in the ocean”. Why does this have to be in the ocean? Isn’t that too specific a location for this work? Same goes for pg. 15.

-Wächtershäuser brought up the idea that polyanionic compounds would bind more favorably to a positively charged surface, which makes sense. How does the number of negative charges quantitatively influence this model?

Reviewer #2: In their manuscript “Positively charged mineral surfaces promoted the accumulation of organic intermediates at the origin of life” Akbari and Palsson study the effects of different properties of potential protocell surfaces/membrane constitutions on the capacity to concentrate organic substances.

Unfortunately, I have to say that after studying the manuscript I was left more confused than I was before reading it. It was incredibly difficult to find a “red line”, because apart from some occasional bright spots, the entire manuscript has no intuitively structured. The erratic narrative style and the mixing of introductory parts and results contribute in particular to this. Assuming that because of my research background, I would be one of the readers who should be particularly interested in the topic, it is to be assumed that the paper in its current form will hardly be read and even less understood. I cannot recommend the manuscript for publication. The Authors are strongly advised to ask their direct peers for critical and honest feedback, always with the aim of making the context understandable to a potential reader.

Due to this reasons and despite my honest efforts, I cannot claim to have really understood the model presented here. That’s why my further feedback relates to a few general issues:

- Especially in the introduction, unsecured hypotheses are often presented as absolute statements. This should be avoided in any case.

- The concept of electroneutrality is upheld in this publication, but there are clear indications that (unlike discussed by the authors) in living cells this is not a hard principle, but rather just a general rule with many exceptions

- Almost every enrichment process (as well as depletion) in living cells works through the constant investment of some form of energy (photons, ATP, synport,..). Where does the assumption come from that this should not also have applied to the first living cells?

**Have the authors made all data and (if applicable) computational code underlying the findings in their manuscript fully available?**

Reviewer #1: Yes

Reviewer #2: Yes

PLOS authors have the option to publish the peer review history of their article (what does this mean?). If published, this will include your full peer review and any attached files.

Reviewer #1: No

Reviewer #2: No
---

## [Editor Report · Decision Letter 1]

11 Jul 2022

Dear Prof. Palsson,

We are pleased to inform you that your manuscript 'Positively charged mineral surfaces promoted the accumulation of organic intermediates at the origin of metabolism' has been provisionally accepted for publication in PLOS Computational Biology.

Best regards,

Markus Ralser

Guest Editor

PLOS Computational Biology

Kiran Patil

Deputy Editor

PLOS Computational Biology

---

## [Editor Report · Acceptance letter]

8 Aug 2022

PCOMPBIOL-D-22-00355R1 

Positively charged mineral surfaces promoted the accumulation of organic intermediates at the origin of metabolism

Dear Dr Palsson,

I am pleased to inform you that your manuscript has been formally accepted for publication in PLOS Computational Biology. Your manuscript is now with our production department and you will be notified of the publication date in due course.

With kind regards,

Olena Szabo
